# Live cell imaging of meiosis in *Arabidopsis thaliana*

**Maria A Prusicki[1], Emma M Keizer[2†], Rik P van Rosmalen[2†], Shinichiro Komaki[1‡], Felix Seifert[1§], Katja Müller[1], Erik Wijnker[3], Christian Fleck[2#\*], Arp Schnittger[1\*]**

[1]Department of Developmental Biology, University of Hamburg, Hamburg, Germany; [2]Department of Agrotechnology and Food Sciences, Wageningen University, Wageningen, The Netherlands; [3]Department of Plant Science, Laboratory of Genetics, Wageningen University and Research, Wageningen, The Netherlands

**\*For correspondence:**
christian.fleck@bsse.ethz.ch (CF); arp.schnittger@uni-hamburg.de (AS)

[†]These authors contributed equally to this work

**Present address:** [‡]Plant Cell Function Laboratory, Nara Institute of Science and Technology, Nara, Japan; [§]cropSeq bioinformatics, Hamburg, Germany; [#]Department of Biosystems Science and Engineering, ETH Zurich, Basel, Switzerland

**Abstract** To follow the dynamics of meiosis in the model plant Arabidopsis, we have established a live cell imaging setup to observe male meiocytes. Our method is based on the concomitant visualization of microtubules (MTs) and a meiotic cohesin subunit that allows following five cellular parameters: cell shape, MT array, nucleus position, nucleolus position, and chromatin condensation. We find that the states of these parameters are not randomly associated and identify 11 cellular states, referred to as landmarks, which occur much more frequently than closely related ones, indicating that they are convergence points during meiotic progression. As a first application of our system, we revisited a previously identified mutant in the meiotic A-type cyclin *TARDY ASYNCHRONOUS MEIOSIS* (*TAM*). Our imaging system enabled us to reveal both qualitatively and quantitatively altered landmarks in *tam*, foremost the formation of previously not recognized ectopic spindle- or phragmoplast-like structures that arise without attachment to chromosomes.
DOI: https://doi.org/10.7554/eLife.42834.001

## Introduction

Meiosis is essential for sexual reproduction by reducing the chromosome number to eventually generate gametes with half the genomic DNA content of the parental plant. Moreover, meiosis is central to the formation of genetic diversity by generating recombination between the homologous chromosomes (homologs) and by randomly selecting either the maternal or paternal homologs to establish a new set of chromosomes in the gametes. Therefore, understanding the molecular mechanisms underlying recombination and chromosome distribution are also of key interest for breeding to modulate meiosis (*Crismani et al., 2013*; *Hand and Koltunow, 2014*; *Lambing and Heckmann, 2018*).

Entry into meiosis is tightly regulated in all organisms. In plants, it involves the reprogramming of somatic fate since plants, in contrast to animals, do not have a germline that is set aside early during embryo development (*Schmidt et al., 2015*). Designated meiocytes have to repress stem cell activity (*Zhao et al., 2017*), and differentiate by adopting a characteristic shape that radically changes during the course of meiosis ultimately resulting in the formation of spores. These spores then differentiate into gametophytes that produce the gametes, which will fuse during fertilization (*Dresselhaus et al., 2016*).

In recent years, our understanding of meiosis in plants has been fostered by genetic approaches, mostly in the model plants *Arabidopsis thaliana*, *Zea mays* and *Oryza sativa*. These studies have identified more than 80 meiotic genes, including those that control entry and progression through meiosis (*Lambing et al., 2017*; *Ma, 2006*; *Mercier et al., 2015*; *Wijnker and Schnittger, 2013*; *Zhou and Pawlowski, 2014*). For instance, mutants in the A1;2 type cyclin *TARDY AYNCHRONOUS MEIOSIS (TAM)* were found to be required for entry into meiosis II (*Cromer et al., 2012*;

**eLife digest** In plant cells, as in other cells, genetic information is stored within structures known as chromosomes. Most of the cells in a plant contain a duplicated set of chromosomes that the plant needs to survive. However, plants also produce some cells known as sex cells that only have a single set of chromosomes. This ensures that, when plants sexually reproduce, a male and female sex cell will fuse together and eventually grow into a new plant that carries a doubled set of chromosomes.

Cells known as meiocytes make sex cells by dividing through a process known as meiosis. Previous studies have identified several genes that regulate meiosis in plants. For example, a gene known as *TAM* is required to make sex cells in a small weed known as *Arabidopsis thaliana*, which is often used as a model plant in research studies.

During meiosis, meiocytes need to copy and move their chromosomes at precisely the right time to ensure that each sex cell they produce has a complete set of chromosomes. Studies of how chromosomes behave during meiosis in plants have so far almost exclusively relied on traditional microscopy techniques that kill the cells in the process of preparing them for imaging. Before being placed under a microscope, the dead cell material is often spread out to make it easier to see the chromosomes. These techniques provide snapshots of meiosis that provide good spatial resolution of chromosome behavior, but information about how chromosomes and other cellular components behave in the course of meiosis is lost.

Prusicki et al. developed a new microscopy approach to observe meiosis in living *A. thaliana* cells. The experiments found that the structure of all the cells changed during meiosis in several distinct stages (referred to as 'landmarks'). Some of these landmarks were absent or happened at a different time in mutant plant cells that lacked the *TAM* gene. As a result, a structure called the spindle that is required to move chromosomes during meiosis formed at the wrong time in the mutant cells.

The findings of Prusicki et al. reveal new insights into the role of *TAM* in meiosis. The next step following on from this work is to use the same approach to study other mutant plants with defects in meiosis and analyze the effects of a changing environment on meiosis.

DOI: https://doi.org/10.7554/eLife.42834.002

*d'Erfurth et al., 2010*; *Magnard et al., 2001*). TAM is a *bona fide* cyclin and builds at least in vitro an active kinase complex with CYCLIN-DEPENDENT KINASEA A;1 (CDKA;1), which is of key importance for both mitosis and meiosis (*Dissmeyer et al., 2009*; *Nowack et al., 2012*; *Wang et al., 2004a*). However, the molecular targets of TAM have not been identified and a mechanistic understanding of its role in meiosis is missing.

Cytological studies of mutants defective in *tam* and other meiotic genes have so far exclusively relied on the analysis of fixed material by cytochemical methods such as chromosome spreads and the immuno-detection of proteins. While these techniques have been, and continue to be, very informative, they capture the underlying cellular dynamics only to a small degree. Importantly, these methods do not allow individual cells to be followed over time. Thus, conclusions about the course of meiocyte development and progression through meiosis have to be deduced from the analysis of different cell populations at different time points.

So far, only two approaches to observe meiosis in real time in plants have been described, revealing details about spindle dynamics and chromosome paring in maize meiocytes. First, the work of Yu et al. and its modification by Nannas et al. used fluorescence microscopy to observe isolated male meiocytes cultured for a maximum of 9 hr in liquid medium (*Nannas et al., 2016*; *Yu et al., 1997*). The method of Nannas et al. combined the DNA dye Syto12 with the expression of β-tubulin fused to CFP, thereby allowing the concomitant observation of chromosomes and MTs. This revealed a spatially asymmetric positioning of the spindle at anaphase I and II, and chromosome-dependent phragmoplast deposition (*Nannas et al., 2016*). The second approach involved imaging entire anthers of maize by exploiting the high depth of field of two-photon microscopy, as earlier proposed by Feijó et al. (*Feijó and Cox, 2001*; *Feijó and Moreno, 2004*; *Sheehan and Pawlowski, 2012*; *Sheehan and Pawlowski, 2009*). This method, which allowed imaging for periods of 24 hr, led to the characterization of three different movements and trajectories followed by the chromosomes during pairing in prophase I (*Sheehan and Pawlowski, 2009*).

The studies in maize relied on visualizing DNA by chemical stains such as Syto12 and DAPI and the power of Arabidopsis as a molecular model, which enables the relatively fast generation of fluorescent reporter lines for different meiotic proteins, has largely not been exploited in combination with live cell imaging of meiosis. A first approach was made by Ingouff et al. who observed methylation changes during Arabidopsis sporogenesis and gametogenesis, albeit without resolving specific meiotic stages (*Ingouff et al., 2017*).

Here, we set out to develop a live cell imaging system for meiosis in Arabidopsis. To this end, we have generated an easy applicable microscopic set up, a combination of meiotic reporter lines covering central aspects of meiosis, and an evaluation system based on morphological characteristics that allowed the quantification of meiotic phases with high temporal resolution. This work gives insights into the robustness of meiocyte differentiation steps and provides important criteria to judge and/or re-evaluate mutants affecting meiosis. As a test case, we have re-analyzed *tam* mutants and find new phenotypic aspects that suggest that TAM is a central factor coordinating the cytoskeleton with nuclear events.

## Results

### Specimen preparation

Live cell imaging can be performed at three general levels and all three have been applied to the analysis of meiosis in multicellular organisms. First, imaging can be performed on isolated cells as for instance seen in the case of mammalian oocytes where confocal microscopy was applied to analyze chromosome segregation: kinetochores could be tracked for over 8 hr, revealing that the bi-oriented attachment of homologs is established after a lengthy try-and-error process (*Kitajima et al., 2011*); microtubule organizing centers (MTOC) and actin elements of the cytoskeleton have been shown to be relevant for spindle formation and correct segregation (*Holubcová et al., 2013*; *Mogessie and Schuh, 2017*; *Schuh and Ellenberg, 2007*), as well as it was confirmed by live cell imaging of fetal mouse oocytes that cohesin establishment is maintained without detectable turnover and that its loss in older oocytes remains uncorrected, leading to formation of aneuploid and non-viable gametes (*Burkhardt et al., 2016*). This approach usually gives very high spatio-temporal resolution since there is no requirement for the laser beam to penetrate surrounding cells and very little laser power can be used reducing photobleaching and phototoxicity. However, since meiocytes in Arabidopsis are very small, that is 20 µm, and difficult to isolate, we did not explore this possibility further. Next, imaging can be carried out in the context of an entire organism, for example in *C. elegans* (*Mullen and Wignall, 2017*; *Rosu and Cohen-Fix, 2017*) with the benefit of perturbing the analyzed cells as little as possible by preparation procedures. However, this set up is limited to small organisms and/or short observation times due to size restrictions and the problem of movement of the sample and thus, cannot be used for Arabidopsis either. Finally, live cell imaging can be performed on isolated organs or tissues that are typically easy to obtain and that provide the appropriate developmental context for analysis of the selected individual cells, for example in mice (*Enguita-Marruedo et al., 2018*), *C. elegans* (*Mlynarczyk-Evans and Villeneuve, 2017*) and *Drosophila melanogaster* (*Głuszek et al., 2015*). As conventional confocal laser scanning microscopes can reach cells up to a depth of 70–100 µm, they are suited to observe the meiocytes in Arabidopsis that are covered by three cell layers in the anthers. Imaging of isolated organs has already been successfully applied to the analysis of organogenesis in the shoot apical meristem (SAM) of Arabidopsis (*Hamant et al., 2014*; *Reddy et al., 2004*; *Reddy and Meyerowitz, 2005*). Since shoots could be maintained for several days without obvious perturbations of development, we decided to adapt and optimize this approach for our purposes.

First, we harvested inflorescences and removed all but one young flower primordium presumably containing meiotic stages as indicated by its round shape and an approximate diameter of 0.4–0.6 mm (*Figure 1*), corresponding to stage 9 of flower development (*Smyth, 1990*). Next, the upper sepal was removed giving access to two of the six anthers since the petals are shorter than the anthers at this floral stage. Finally, the bud along with the pedicel and a few millimeters of the stem was embedded into Arabidopsis Apex Culture Medium (ACM) and stabilized with a drop of agarose (*Figure 1A,B*). In agreement with the previous analysis of the SAM, we found that the flower buds

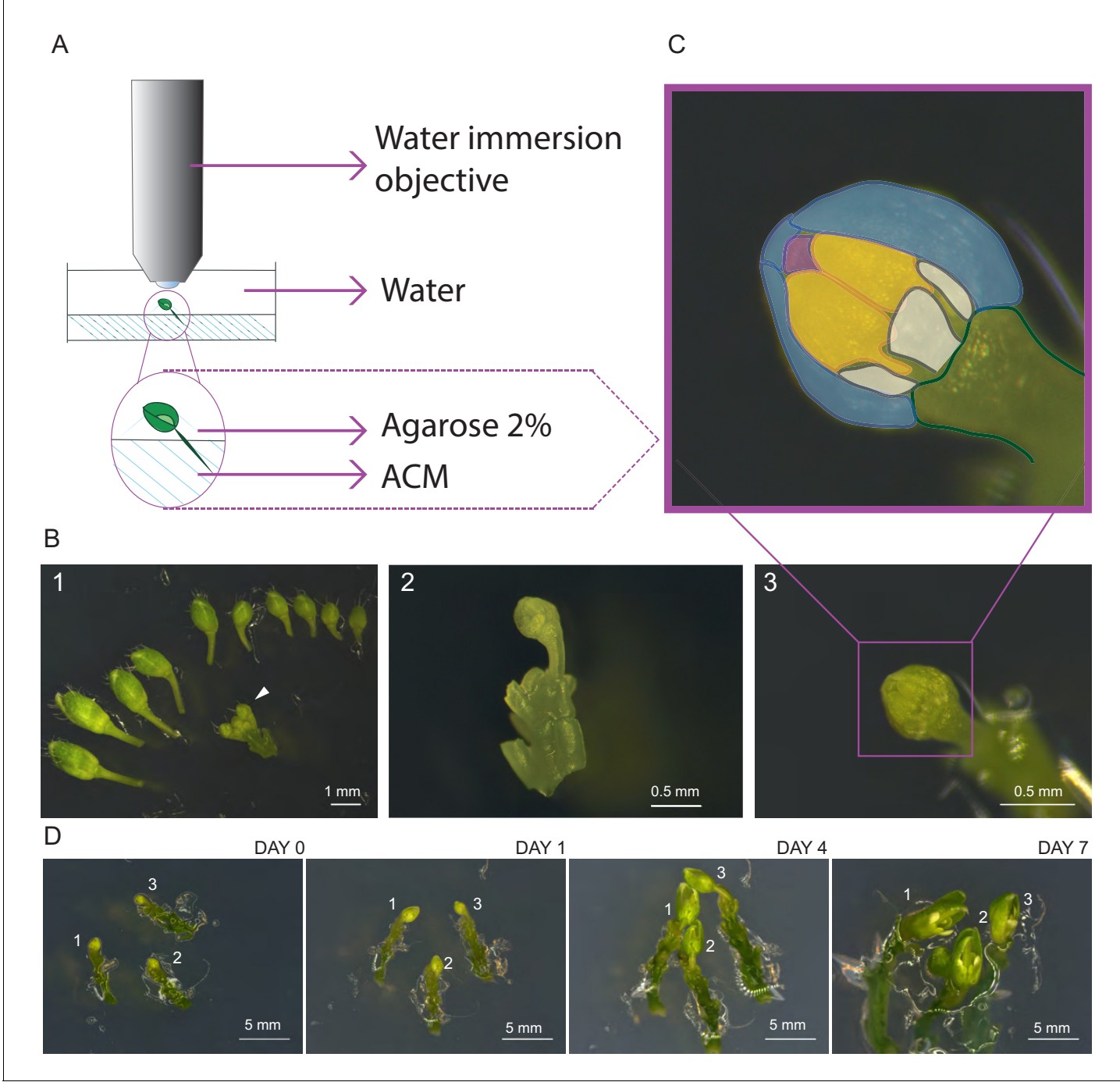

**Figure 1.** Establishment of life cell imaging of male meiocytes. (A) Microscope set-up for live cell imaging. An isolated flower bud is mounted on a small petri dish in ACM medium, stabilized with a drop of 2% agarose and submerged in sterile water. The objective is directly dipped into the water. (B) Steps of sample preparation. A flower bud of 0.4–0.6 mm length is selected (white arrow-head, B1). The upper-most sepal of this flower and all other flowers are removed from the stem (B2). The flower is anchored in the medium with the anthers exposed to the top (B3). (C) Magnification of the sample shown in B3. The two exposed anthers are highlighted in yellow, petals in white, the three remaining sepals in blue, and the tip of the stigma in pink. (D) Flower buds can be kept alive and growing for up to 1 week. Numbers 1, 2 and 3 mark the same flower buds over time.
DOI: https://doi.org/10.7554/eLife.42834.003

The following figure supplement is available for figure 1:

**Figure supplement 1.** Pollen undergoes a further cell division after imaging.
DOI: https://doi.org/10.7554/eLife.42834.004

stayed alive on the ACM medium for up to seven days during which all the flower organs expanded, revealing that cells were able to undergo several divisions on the medium (*Figure 1C*).

Imaging was performed with an up-right confocal laser scanning microscope equipped with a water immersion objective. The entire flower bud was submerged in water and the objective was brought into direct contact with the sample (*Figure 1A*). During image acquisition the temperature was kept constant at 21°C.

To test viability of the sample after image acquisition, we transferred flower buds, which were imaged for 24 or 48 hr, onto new ACM medium and allowed them to grow for 3 days. Concomitant with the growth of the entire flower bud, we found that the microspores derived from imaged meiocytes underwent at least one more cell division giving rise to bi-cellular pollen as revealed by DAPI staining (*Figure 1—figure supplement 1*), confirming that meiocytes are still alive after imaging.

## Establishment of meiotic reporter lines

A generic set up for imaging of cell divisions includes a reporter that highlights DNA/chromatin coupled with a marker for cytoskeletal components, usually MTs, so that chromosome and spindle behavior can be visualized (*Nannas et al., 2016*; *Peirson et al., 1997*). Since fusions of histones with fluorescent proteins have often been applied for this purpose, we first scanned through previously generated transgenic lines expressing different histone variants fused to fluorescent proteins, such H2B. However, while these labeled histones clearly marked DNA in somatic cells, the signal was often fuzzy in meiosis. Moreover, since all or most cells in an anther produced these fusion proteins, the identification of meiocytes was sometimes difficult, especially at early stages of meiosis when chromosomes are not condensed and meiocytes cannot easily be recognized by their size and shape. Therefore, we aimed for a meiosis-specific gene and generated a *GFP* fusion to *REC8*, the alpha kleisin subunit of the cohesin complex, also known as *SYN1* or *DIF1* in Arabidopsis (*Bai, 1999*).

Cohesin is key for chromosome segregation and its step-wise removal allows the segregation of homologous chromosomes in meiosis I, followed by separation of sister chromatids in meiosis II. In addition, cohesin is required for recombination and repair of DNA double-strand breaks resulting in a highly pleiotropic phenotype that leads to almost complete sterility of *rec8* mutant plants (*Bai, 1999*). Expression of our genomic *PRO_{REC8}:REC8:GFP* reporter in a homozygous *rec8* mutant background completely restored fertility of these plants and analysis of chromosome spreads confirmed that chromosome segregation is indistinguishable from the WT (*Figure 2—figure supplement 1*).

REC8 replaces the mitotic RAD21 in meiosis and is hence highly specific for meiocytes in all species analyzed so far (*Nasmyth, 2001*). Consistent with previous immuno-localization studies, we found that the GFP signal of our functional reporter line was only present in meiocytes providing a straightforward way to identify microspore mother cells (*Figure 2*).

Moreover, the REC8 reporter allowed us to estimate the sensitivity of our imaging procedure. While REC8 is removed from chromosome arms at the end of meiosis I to allow the resolution of cross-overs, a small fraction remains at the centromeres to maintain sister chromatid cohesion. The detection of the centromeric fraction of REC8 has been challenging in immuno-localization studies. When we followed the first meiotic division, we observed the remaining REC8 at centromeres indicating that our live cell imaging system is highly sensitive (*Figure 2B*, *Video 1*).

Next, we combined the *PRO_{REC8}:REC8:GFP* with *PRO_{RPS5A}:TagRFP:TUB4* or *PRO_{RPS5A}:TagRFP:TUA5* that label MTs and hence permit observation of the cell shape and the formation of the spindles. The resulting double reporter line is referred to as *Kleisin IN Green microtuBules In ReD (KING-BIRD)* in the following. Plants carrying the double constructs, as well as plants expressing only the *PRO_{RPS5A}:TagRFP:TUB4* or *PRO_{RPS5A}:TagRFP:TUA5* reporter did not have meiotic defects and seed production was as in the WT (*Figure 2—figure supplement 1*).

The separated excitation and emission spectra of the two different fluorochromes permitted faithful and concomitant detection of both, the REC8 and the tubulin reporter.

An important question was how many frames per time interval should be taken. Due to photobleaching as well as potential photo-toxicity, a sampling rate of several frames per minute was not compatible with capturing the entire meiotic program. Based on a several test samples as well as previously published time courses (*Armstrong et al., 2003*; *Sanchez-Moran et al., 2007*; *Stronghill et al., 2014*; *Yu et al., 1997*), we decided to acquire one frame every 3 to 15 min, so that

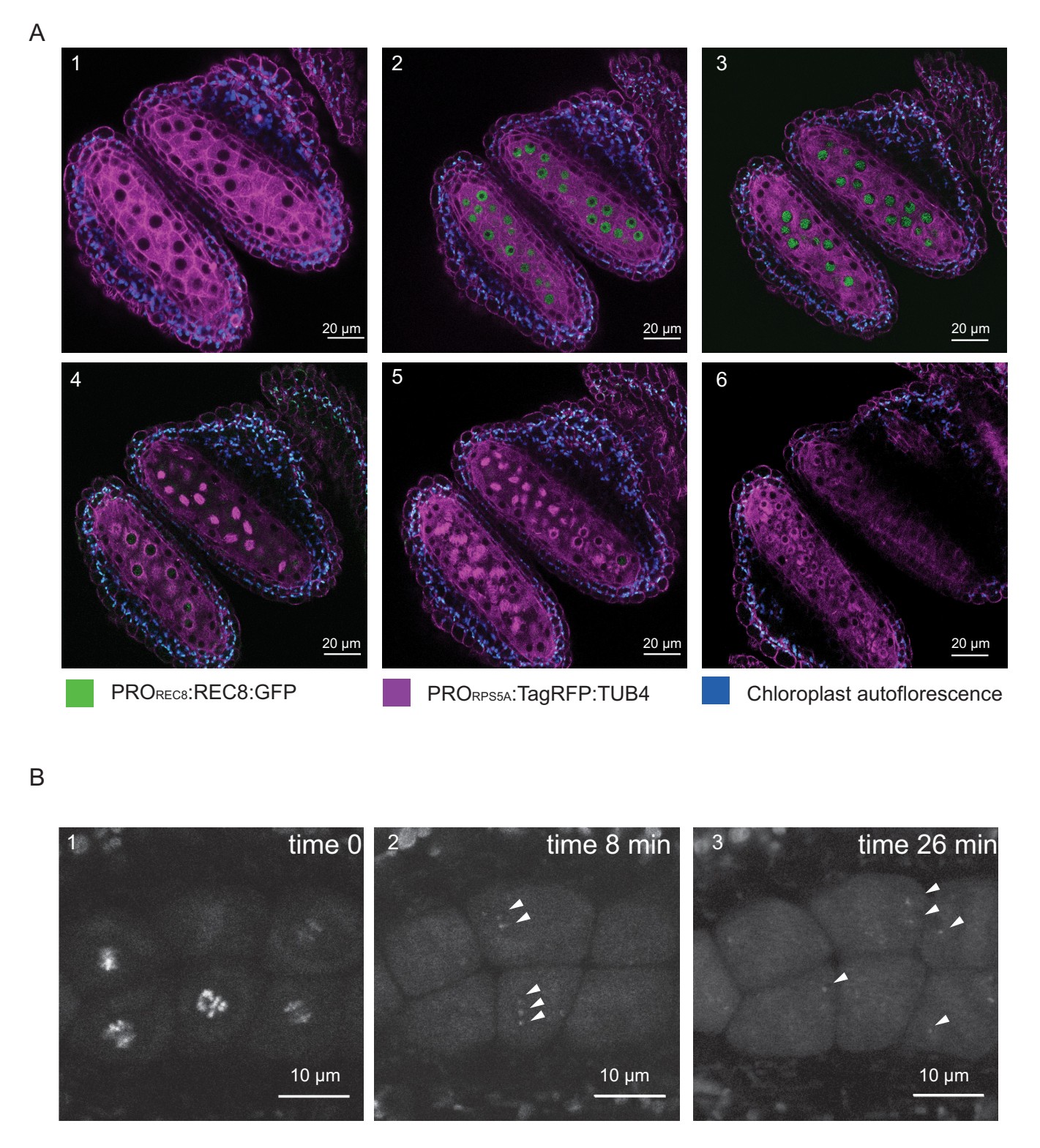

**Figure 2.** REC8 and TUB4 localization. (**A**) Cross section of two neighboring anthers of the *KINGBIRD* line expressing *PRO$_{REC8}$:REC8:GFP* and *PRO$_{RPS5A}$:TagRFP:TUB4* in wild-type background. A1: premeiosis; A2: leptotene; A3: zygotene; A4: diplotene in the lower anther and metaphase I in the upper anther; A5: telophase I in the lower anther and late prophase II-metaphase II transition in the upper anther; A6: tetrad stage. (**B**) REC8:GFP localization after metaphase I (B1) in a *PRO$_{REC8}$:REC8:GFP* plant. The white arrowheads in B2 and B3 indicate centromeric REC8.
*Figure 2 continued on next page*

*Figure 2 continued*

DOI: https://doi.org/10.7554/eLife.42834.005

The following figure supplement is available for figure 2:

**Figure supplement 1.** Functionality of reporter lines used in this study.

DOI: https://doi.org/10.7554/eLife.42834.006

even the shortest phases such a metaphase I and II, could be captured while photo-bleaching was reduced to a minimum.

## A meiotic landmark system

Meiosis is classically apportioned into nine phases: prophase I, metaphase I, anaphase I, telophase I, interkinesis, prophase II, metaphase II, anaphase II and telophase II. Due to the dramatic changes in chromatin structure and the dynamics of chromosomes, and to its prolonged duration, prophase I is divided into the five sub-phases, that is leptotene, zygotene, pachytene, diplotene and diakinesis. These phases have been derived from observation of fixed material and chromosome spreads, leading to definitions mainly based on chromosome configurations, for example pachytene is defined by the presence of fully synapsed chromosomes.

Using the *KINGBIRD* reporter line, we were able to distinguish five parameters of meiocytes: cell shape, MT array, nucleus position, nucleolus position, and chromosome configurations (condensation and pairing/synapsis) (*Figure 3A*). Some of these parameters could be identified as a direct read-out of the reporters, for example cell shape and MT array are visualized by tubulin while chromosome configurations are revealed by REC8. Other parameters could be indirectly determined as for the nucleus position, which is defined by the absence of MTs, or the nucleolus position, which corresponds to the area of the nucleoplasm where REC8 is present. Each parameter can adopt different states, which have a distinct order. For instance, we observed that cell shape always changed from rectangular to trapezoidal, then to oval, to circular, to triangular and finally giving rise to tetrads composed of four triangular cells, among which typically only three can be identified on the same focal plane (*Figure 3A*).

The MT array is the parameter with the largest number of states. At the onset of meiosis, MTs are homogeneously distributed in the cytoplasm (state 1) and then progressively form an arc structure, hereby named *half moon*, which develops on one side of the nucleus, which moves at the same time towards a corner of the cell (states from 2 to 4). The *half moon* then develops into a full-moon-like structure surrounding the nucleus (state 5) and contracts to form a pre-spindle similar to what is observed in mitosis (state 6). At the moment of nuclear envelope break down, the pre-spindle is disrupted (state 7) and MTs rearrange to form the first meiotic spindle (states 8 and 9). States 10 and 11 are present during the transition between meiosis I and meiosis II, MTs reorganize radially around the two new nuclei while the central MTs broaden their disposition forming a phragmoplast-like structure. The second meiotic division resembles the first meiosis, with the formation of two pre-spindles (state 12) followed by two spindles (state 13) and phragmoplast-like structures

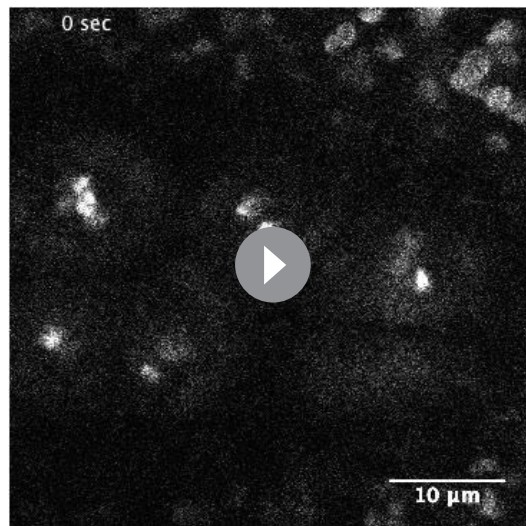

**Video 1.** Detection of $PRO_{REC8}$:REC8:GFP at metaphase I –anaphase I transition. This movie focuses on five male meiocytes at metaphase I. The $PRO_{REC8}$:REC8:GFP signal (in white) is seen on highly condensed chromosomes. With the onset of anaphase I, the remaining $PRO_{REC8}$:REC8:GFP can be seen at the centromere areas of homologs being pulled to opposite cell poles (white arrowheads at the movie reply).

DOI: https://doi.org/10.7554/eLife.42834.007

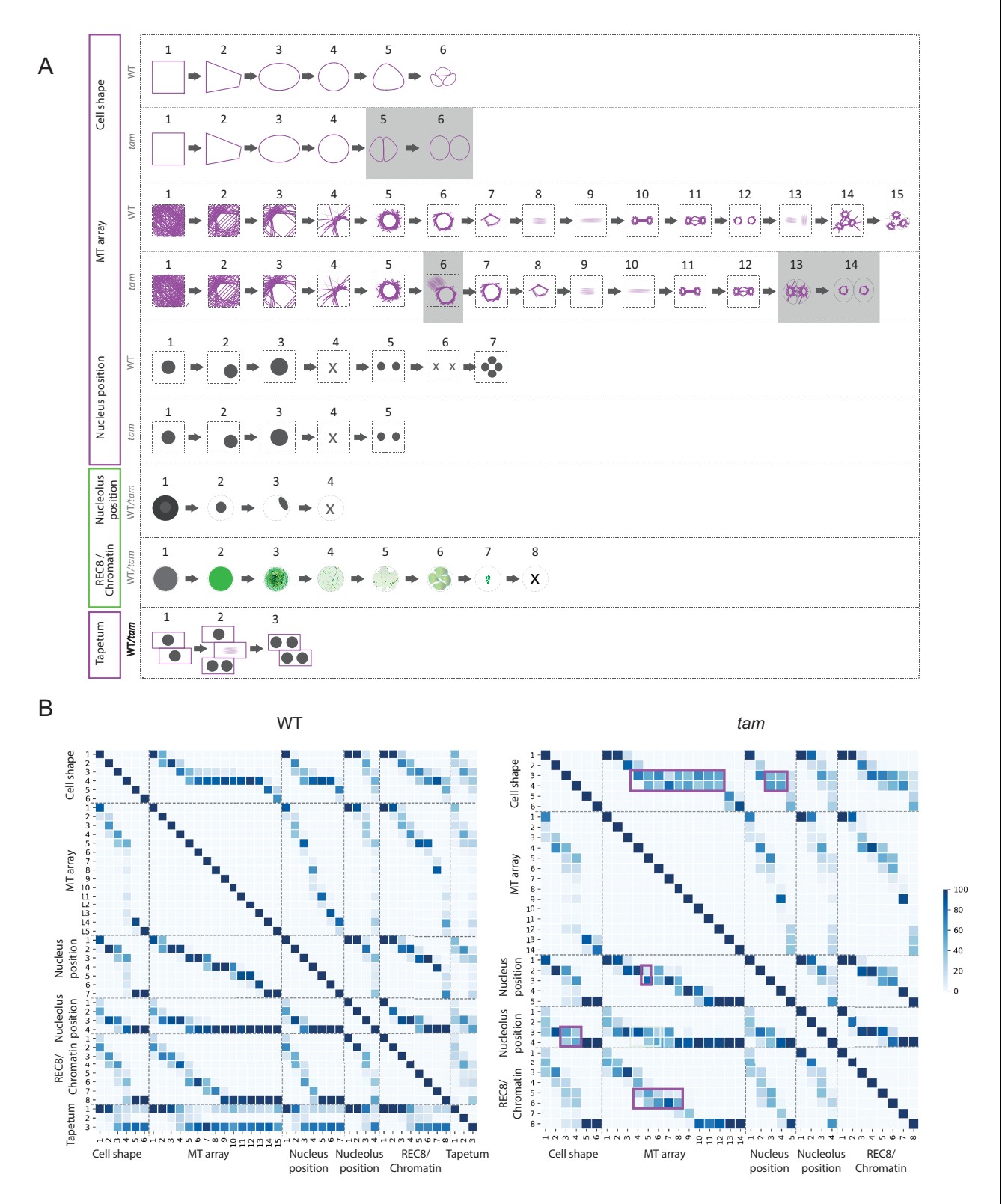

**Figure 3.** Identification of parameter states of male meiocytes. (A) Schematic representation of the different states of the five parameters observed during meiotic progression in the WT and *tam*. While nucleolus position, REC8/chromatin and tapetum cells do not change their patterns; cell shape and MT array adopt additional states in *tam*. Consistent with a premature exit from meiosis, parameter states of the cell shape, nucleus, and MT associated with the second meiotic division are missing in *tam*. (B) Heat-maps of the co-occurrence of the different parameter states in the WT (on the

*Figure 3 continued on next page*

*Figure 3 continued*

left) and *tam* (on the right). The darker the blue color, the higher is the frequency of co-appearance of two parameter states. Tapetum cell states were only included in the WT analysis and found to be not very tightly correlated with any of the other parameters. The magenta rectangle highlights relationships that become less stringent in *tam* when compared to the WT. Numbers refer to the scheme in A.
DOI: https://doi.org/10.7554/eLife.42834.008

(state 14) until the cells undergo cytokinesis and form a tetrad (state 15).

Nucleus and nucleolus are characterized by changes of their positions. At the beginning of meiosis, the nucleus is centrally located (state1). Around the time of the formation of the MT half moon structure, it then moves to one side of the cell (state 2) and at state 3 the nucleus is back to the center of cell. This state is distinguishable from state 1 due to the size of the nucleus, which is now enlarged. During the two metaphases (states 4 and 6), the nuclear structure disappears, to reappear as two and four smaller nuclei at the following states 5 and 7, respectively.

The nucleolus becomes visible at the onset of meiosis, together with the accumulation of REC8. It is initially positioned in the center of the nucleus (state 2) and moves to the nuclear periphery during the progression of prophase (state 3). At late prophase it disappears (state 4). Since the nucleolus is only visible when REC8 is expressed, its reappearance after chromosome segregation could not be noticed.

The last parameter obtained is the localization of REC8, which correlates with chromatin conformation during the first meiotic division. The states identified resemble the localization pattern of REC8 described by immunolocalization experiments (*Cai et al., 2003*). At first, a diffuse fluorescence signal accumulates in the nucleoplasm (state 2), which then condenses to form threads that thicken over time (states 3 and 4) indicating the pairing of chromosomes. At state 5, the REC8 signal becomes fainter consistent with the onset of REC8 removal from chromosomes by the prophase pathway (*Yang et al., 2019*). Soon, the chromosome threads are unrecognizable but a faint diffuse REC8 staining persists (state 6) until small, distinct dots at state 7 are observed representing fully condensed chromosomes. State 7 is followed by the almost complete disappearance of REC8 (state 8), corresponding to the onset of anaphase I.

Importantly, our markers recapitulated previously described changes in nucleolus position, REC8 localization and MT cytoskeleton organization, corroborating that our imaging system does not disturb the overall progression of meiosis (*Peirson et al., 1997*; *Stronghill et al., 2014*; *Wang et al., 2004b*).

Analyzing a first set of movies gave rise to the hypothesis that some of the parameter states are connected, for example the nucleolus apparently dissolves only after the nucleus has moved to one side of the meiocyte and returned to a central position. To assess the nature of these associations, we analyzed a subset of cells (n = 169 from 35 anthers) assigning a combination of numbers that represents each parameter state at every time point when a frame was taken, for example 1-1-1-2-2 describes a meiocyte that is rectangular in shape, has an evenly distributed MT array, a centrally located nucleus with a centrally located nucleolus, and with uncondensed, yet not paired chromosomes. In the following, we call a combination of all five parameter states a *cellular state*.

A subsequent analysis of 10,671 time points allowed us to judge which parameter states occur together and in which frequency (*Figure 3B*). By this method we could confirm that certain parameter states indeed co-occur with each other in a highly specific manner, for example cell shape state 1 is only found to be associated with nucleus position state 1 (*Figure 3B*, WT heatmap), while others parameter states are only more loosely associated.

This analysis also revealed that out of the more than 20,000 possible cellular states only 101 were actually present in our data set (*Figure 4—source data 1*). However, their frequencies were distributed in a very broad range (from 0.01% to 21.14% of the total number of observations). Hence, the importance of a certain cellular state cannot be deduced from the absolute frequency of occurrence since this is highly biased by the duration of the respective state, that is combinations of parameters that depict long phases such as pachytene are present in higher number of time points than combinations depicting short phases, for example metaphase I. To identify biologically distinct cellular states from the observed data, we defined a local or *neighboring score*, which quantifies the occurrence of a certain cellular state compared to its neighboring states.

A neighboring state was defined as a cellular state that is one transition away (−1 or +1) for at least one, but at most two, parameter states compared to the cellular state analyzed. With this, 2-4-2-3-4, for example, is a neighbor of 2-3-2-3-4 and of 3-3-2-3-4, but not of the cellular state 2-2-2-3-4 and not 3-3-2-3-3 (*Figure 4—source data 1*). Notably, we only took states into account that were actually observed. The neighboring score was then compared with the subset of *neighboring states*, to find the predominant state among the surrounding states, and is defined as:

$$\text{Score} = \frac{\text{count(state)} - \text{mean(count(neighboring states))}}{\text{std(count(neighboring states))}}$$

where *counts* refers to the number of times a certain state is observed in the data, and *std* refers to the standard deviation. This analysis revealed 11 clearly distinct cellular states that differed from their neighbors with a score higher than one, denoting that they occurred at least one standard deviation more frequent than the mean of the neighboring stages (*Figure 4*).

These 11 cellular states (A1-A11) are henceforth called *meiotic landmark states* or *landmarks* (*Figure 4* and *Figure 5*). The states between landmarks are defined as *transition states*, and often represent alternative routes to the next landmark (*Figure 4*), for example the cell shape may first change from rectangular to trapezoidal and then the nucleus moves from a center position to a position at the side of the cell, or the nucleus moves first and then the cell shape changes. However, the nucleus is finally always located at the smaller side of the trapezoidal cell defining the new landmark state. The results of the neighboring score analysis were reproduced and confirmed by bootstrapping (*Figure 4—source data 2*).

Taken together, we conclude that cellular differentiation steps of meiosis can be variable but then converge on distinct cellular states, the landmarks. The qualitative assortment of the landmarks, possibly their order as well as their duration and the degree of variability (transition state number and duration), represent a new system to describe meiosis.

## The case of the nuclear envelope breakdown

The break-down of the nuclear envelope in diplotene is an important hallmark of meiosis (*Wijnker and Schnittger, 2013*). We also could clearly observe the breakdown in our live cell imaging system although, due to its rapid progression, it was only captured in 22 out of 10,671 analyzed time points with a sampling interval of one frame every 10 min (*Figure 6* and *Video 2*). Nonetheless, the nuclear envelope break-down is not included in a landmark state since it appeared to be only loosely connected with the other parameter states, for example the cell shape can be oval or round, and the chromatin can be at different condensation levels when the nuclear envelope breaks down (*Figure 6*). Thus, although very distinct when looking at MT conformation (i.e. state 7, collapse of pre-spindle *Figure 3A*), a clearly defined landmark state corresponding to nuclear envelope break-down was not reached with the parameters analyzed.

We could also clearly observe other short-lived phases such as diakinesis, anaphase I, prophase II and anaphase II. However, due to their unexpected high variation in terms of association with the here analyzed parameter states, these phases, like nuclear envelope breakdown were also not designated as landmarks.

## Correlation between meiocyte and tapetum differentiation

Our sample preparation, which keeps anthers intact, also provided the possibility to follow the differentiation of the tissues surrounding the meiocytes, especially the tapetum cells. These are in direct contact with the meiocytes and are thought to nourish and support the meiocytes and spores (*Pacini et al., 1985*). A key feature of tapetum cells in many plant species, including Arabidopsis, is that they become poly-nucleated through endomitosis, that is a cell cycle variant in which cytokinesis is skipped (*Jakoby and Schnittger, 2004*). The poly-nuclearization of tapetum cells was clearly visible in our *KINGBIRD* line (*Video 3*, from minute 980 to minute 1207), possibly representing a sixth cell parameter next to the five meiotic parameters presented above (*Figure 3A*). Notably, tapetum cell differentiation was previously suggested as a criterion to judge stages of meiosis (*Stronghill et al., 2014*; *Wang et al., 2004b*). We observed that polynucleated tapetum cells are not found before A4/zygotene and conversely, when all tapetum cells are poly-nucleated, meiosis has progressed into A7/diplotene. However, endomitosis only poorly correlated with any of the

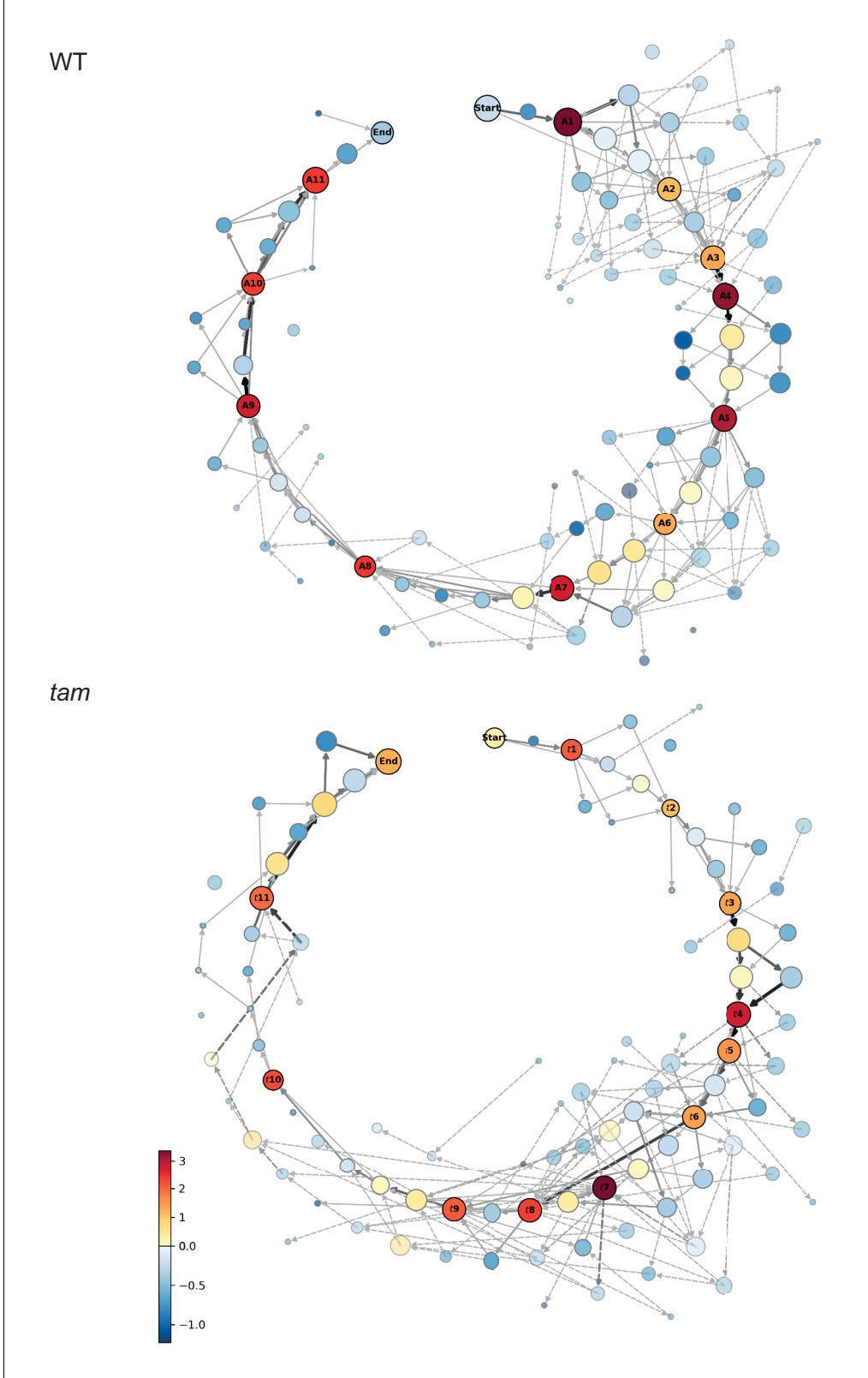

**Figure 4.** Matrix of cellular states. The two schemes represent the matrix of the observed cellular states and how they relate to each other, in the WT (upper scheme) and *tam* background (lower scheme). Each circle represents a cellular state as a function of the combination of the observed parameters states. The area of the circle indicates the number of observations of that particular combination while the color indicates its neighboring score value; the warmer the color the higher is the neighboring value. The circles with a name and marked by a dark outline are the selected landmark
*Figure 4 continued on next page*

*Figure 4 continued*

states. The remaining circles are defined as transition states. The arrows represent all the direct transitions between cellular states observed within the data set, the thicker the arrows, the higher the number of observations. The dotted lines are transitions that do not fit the landmark scheme for example a trapezoidal cell (cell shape state 2) paired to an half moon initial state (MT array state number 3), yet with a centrally placed nucleus (state number 1). Most of these states are observed very rarely (2.8% in total).

DOI: https://doi.org/10.7554/eLife.42834.009

The following source data is available for figure 4:

**Source data 1.** Cellular states of WT plants.
DOI: https://doi.org/10.7554/eLife.42834.010
**Source data 2.** Bootstrapping for WT landmark scoring.
DOI: https://doi.org/10.7554/eLife.42834.011
**Source data 3.** Cellular states of tam.
DOI: https://doi.org/10.7554/eLife.42834.012
**Source data 4.** Bootstrapping for tam landmark scoring.
DOI: https://doi.org/10.7554/eLife.42834.013

meiotic stages between A4 and A7 (*Figure 3B*) and hence, was not incorporated into the landmark system. In turn, we conclude that the meiotic progression and tapetum cell differentiation are not tightly correlated.

## A landmark analysis can be performed with one marker only but is less informative

One obvious future experiment is to combine the here-developed landmark system with additional reporter lines for meiotic regulators, for example ZYP1 and ASY3 (*Yang et al., 2019*). However, green fluorescent proteins (GFP and possibly mNEONgreen) are still by far the most powerful reporters due to their high quantum yield and photostability, especially for poorly expressed genes. Hence, we were interested in which landmarks could be revealed using only one color, that is either $PRO_{REC8}$:REC8:GFP or $PRO_{RPS5A}$:TagRFP:TUB4 alone leaving the second color, RFP and GFP, respectively, for labeling another protein of interest.

$PRO_{REC8}$:REC8:GFP allows the observation of chromatin condensation levels and of nucleolus position, while $PRO_{RPS5A}$:TagRFP:TUB4 reveals cell shape, nucleus position and MT array (*Figure 3A*). As expected, excluding some parameters by relying on only one reporter resulted in a reduced number of observed cellular states: 52 for $PRO_{RPS5A}$:TagRFP:TUB4 and 14 for $PRO_{REC8}$:REC8:GFP, compared with the 101 states identified by analyzing the KINGBIRD line; (*Figure 5—source data 1*). Ultimately, this led to a lower number of landmarks with a neighboring score higher than 1 in comparison to the analysis with two reporters (*Figure 5* and *Figure 5—source data 1*). Analysis of $PRO_{REC8}$:REC8:GFP by itself delivered landmarks A1, A5 and A7 while $PRO_{RPS5A}$:TagRFP:TUB4 revealed landmarks A1, A3 and A7 to A11 (*Figure 5*). Notably, the landmarks revealed by the KINGBIRD line are not simply the addition of the landmarks unraveled by $PRO_{REC8}$:REC8:GFP and $PRO_{RPS5A}$:TagRFP:TUB4. A2, A4, and A6 only appeared as landmarks when both reporters are used indicating the added value of using multiple reporters.

Among the two reporters, $PRO_{RPS5A}$: TagRFP:TUB4 turned out to be the most informative when used alone due to the fact that its accumulation pattern covers the complete division from pre-meiotic phases to tetrad stage, and that MT behavior is very distinct in meiosis. Thus, $PRO_{RPS5A}$:TagRFP:TUB4 can be used as a landmark reporter alone and is especially useful for stages after meiosis I.

## Time course of meiosis in male meiocytes

The traditional definition of meiosis, mostly relying on chromosome spreads, and the here-established landmark system by live cell imaging are based on different parameters and aspects of meiosis. Nonetheless, we could, at least roughly, align our landmark-based classification with the traditional definition of meiosis (*Figure 5*). We could attribute A1 to an interval between S-phase and early leptotene based on the starting expression of REC8 and its loading onto chromosome arms (*Cai et al., 2003*). A2 could be associated to late leptotene when chromosomes appear as thin threads, as recognized in our case by the REC8 reporter. In addition, the nucleolus, which can be

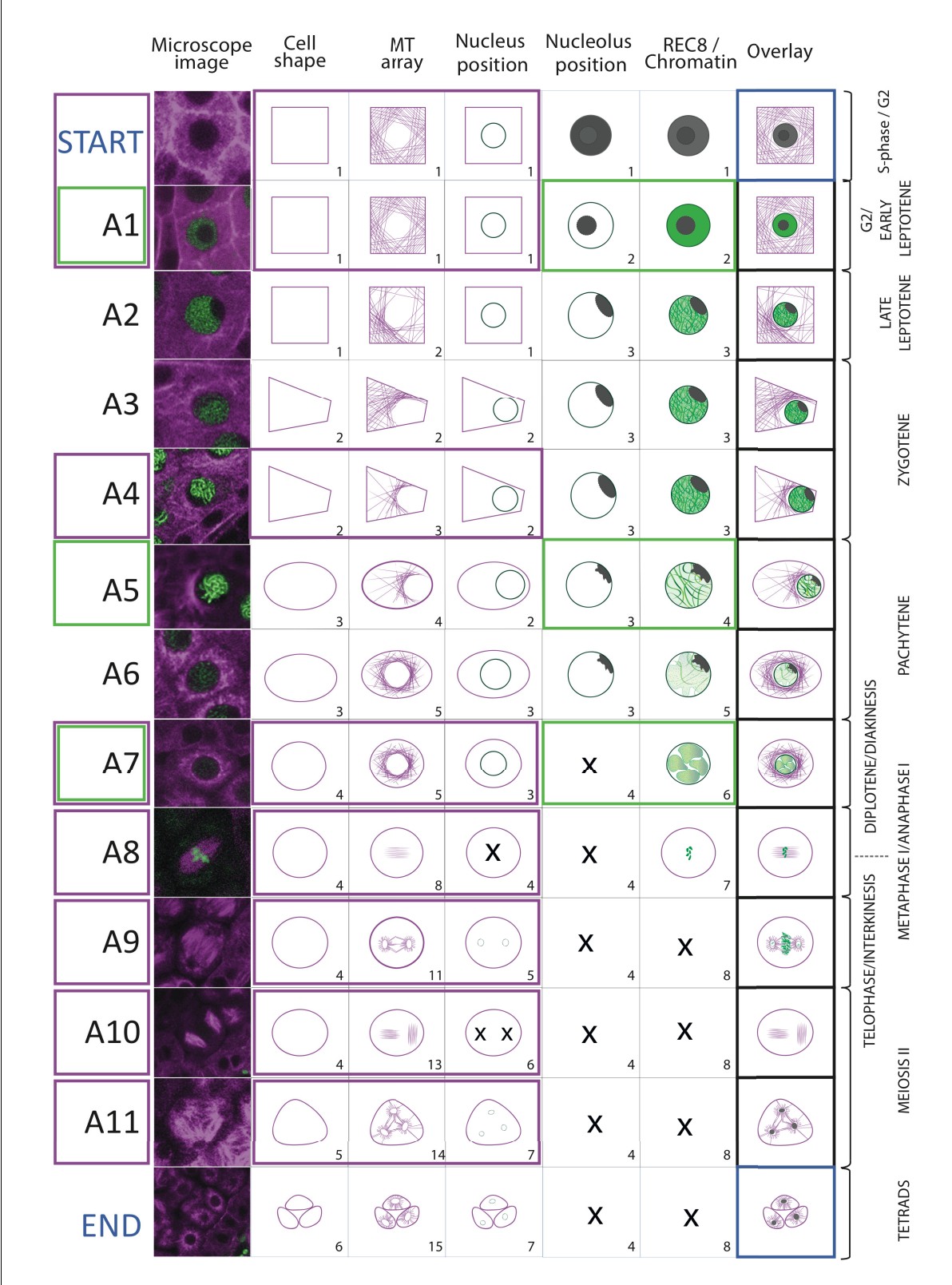

**Figure 5.** Landmark scheme Illustration of the 11 meiotic landmarks. (**A1-A11**) identified by the neighboring score in WT male meiosis. The first column provides a microscopy picture of meiocytes depicting each cellular state. The state of each parameter is separately shown in the following columns, the right-most column (overlay) displays their combination. On the right side, the classical stages of meiosis are roughly assigned to each landmark. The magenta and green frames identify the landmarks obtained by the analysis of TagRFP:TUB4 and REC8 separately.

*Figure 5 continued on next page*

*Figure 5 continued*

DOI: https://doi.org/10.7554/eLife.42834.014

The following source data is available for figure 5:

**Source data 1.** Cellular states revealed by 1 reporter.

DOI: https://doi.org/10.7554/eLife.42834.015

detected by our system by the absence of REC8, moves to the periphery of the nucleus, which is

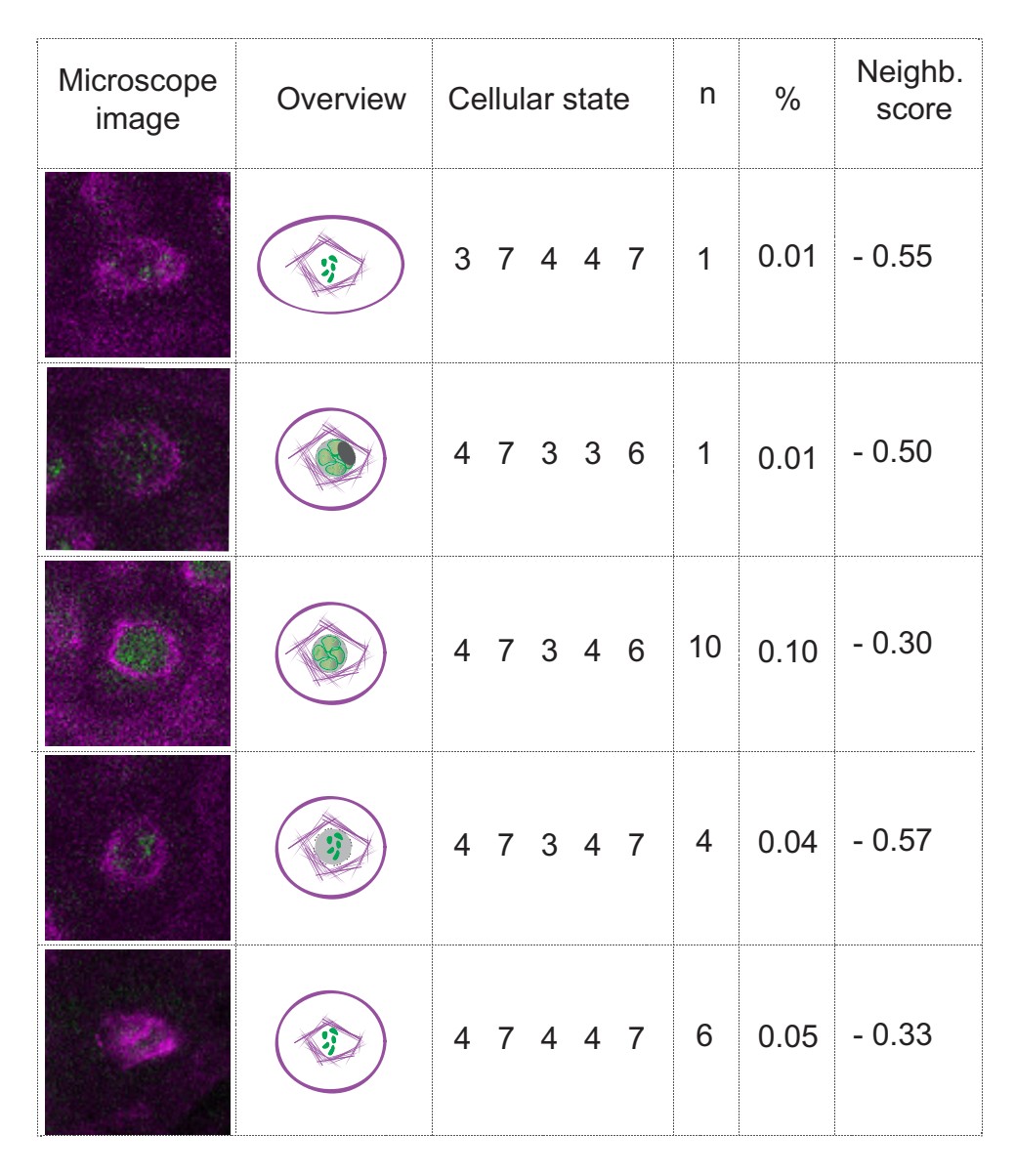

| Microscope image | Overview | Cellular state | n | % | Neighb. score |
|---|---|---|---|---|---|
| | | 3 7 4 4 7 | 1 | 0.01 | - 0.55 |
| | | 4 7 3 3 6 | 1 | 0.01 | - 0.50 |
| | | 4 7 3 4 6 | 10 | 0.10 | - 0.30 |
| | | 4 7 3 4 7 | 4 | 0.04 | - 0.57 |
| | | 4 7 4 4 7 | 6 | 0.05 | - 0.33 |

**Figure 6.** The case of nuclear envelope breakdown. Table illustrating the different parameter states and the corresponding microscope images at the moment of nuclear envelope breakdown in WT plants. Even when the breakdown can be seen (the low number of observations is due to the short duration of the phenomenon), there is high variability of the combinations of parameter states that depict this moment. Hence, the neighboring scores are below zero, precluding the inclusion of the nuclear envelope breakdown as a landmark in this analysis.

DOI: https://doi.org/10.7554/eLife.42834.016

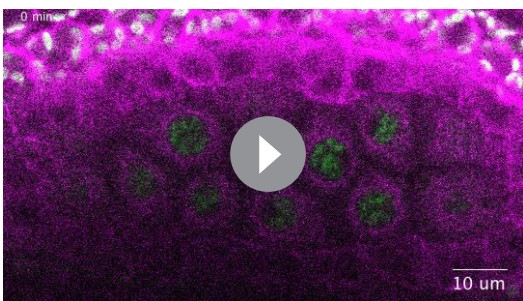

**Video 2.** NE breakdown in WT male meiocytes at stages from diplotene to metaphase. The instance of nuclear envelope breakdown can be seen for the majority of the cells at minute 75 and for the remaining cells at minute 80. Tubulin is highlighted in magenta; chromosomes and REC8 are in green. The movie has been acquired with 5 min interval time.
DOI: https://doi.org/10.7554/eLife.42834.017

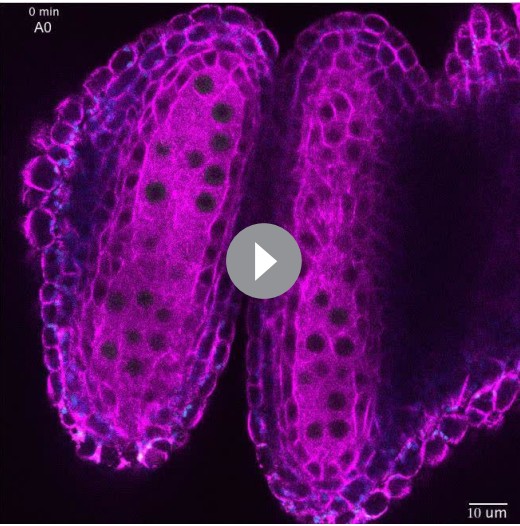

**Video 3.** Complete meiotic progression in WT. Progression of meiosis in two anthers of a WT flower. Tubulin (RFP) is highlighted in magenta, chromosomes are marked by REC8 in green (GFP), chloroplasts (autofluorescence) are blue. The meiocytes, localized in the central areas of the pollen sacs, reside in a pre-meiotic stage at the beginning of the movie, and undergo a complete meiotic program with the first and the second meiotic division until the formation of tetrads. On the top left corner, there is an indication of the landmark crossed. Time is expressed in minutes; the interval between image acquisition is 10 min, with the exception of 1 time point (7 min interval between time 1130 and 1137); time 0 corresponds to the start of image acquisition, and not to the start of meiosis.
DOI: https://doi.org/10.7554/eLife.42834.018

marked by REC8, as described for leptotene cells (*Armstrong and Jones, 2003*; *Ross et al., 1996*; *Stronghill et al., 2014*). A3 and A4 fall into zygotene as zygotene cells have previously been found to have the majority of organelles and MTs localized on only one side of the nucleus (*Armstrong and Jones, 2003*; *Peirson et al., 1997*; *Ross et al., 1996*; *Stronghill et al., 2014*). Additionally, we observe a thickening of chromosome threads that would be consistent with the formation of the synaptonemal complex, which starts to be formed in zygotene (*Higgins et al., 2005*). Pachytene is characterized by the complete synapsis of homologous chromosomes and the repositioning of the nucleus into the central area of the cell (*Armstrong et al., 2003*; *Armstrong and Jones, 2003*; *Ross et al., 1996*). Therefore we could link landmarks A5 and A6 to pachytene. A7 is characterized by a diffuse signal of REC8, which is consistent with the release of synapsis in diplotene (*Cai et al., 2003*; *Ross et al., 1996*). A8 can be identified by the formation of a single spindle and five highly condensed chromosomes that align in the metaphase plate, hence representing metaphase I. In A9 two nuclei appear, still connected by MTs, revealing that this stage is telophase I/interkinesis, followed by the formation of two spindles in A10, which is metaphase II. Finally A11, where three to four distinct nuclei can be detected without being separated by cell walls, represents telophase II.

The assignment of landmarks to the classical stages allowed us then to compare the length of meiotic phases in our live cell imaging approach with the previously performed time course experiments in which the length of meiosis has been estimated by pulse-chase experiments applying either the modified thymine analog 5-bromo-2'-deoxyuridine (BrdU) or 5-ethynyl-2'-deoxyuridine (EdU) to plants. After a given amount of time, meiotic spreads were prepared and tested for the appearance of these analogs in meiotic chromosome configurations. In these experiments, male meiosis in Arabidopsis was judged to last from G2 onwards approximately 32 to 33 hr with leptotene spanning between 6 and 7 hr, zygotene and pachytene together lasting between 12 and 16 hr. Notably, these previous pulse-chase experiments were not able to resolve stages after diplotene and the rest of meiosis (from diplotene onwards) were estimated to approximately persist for 3 hr (*Armstrong et al., 2003*; *Sanchez-Moran et al., 2007*; *Stronghill et al., 2014*) (*Figure 7*).

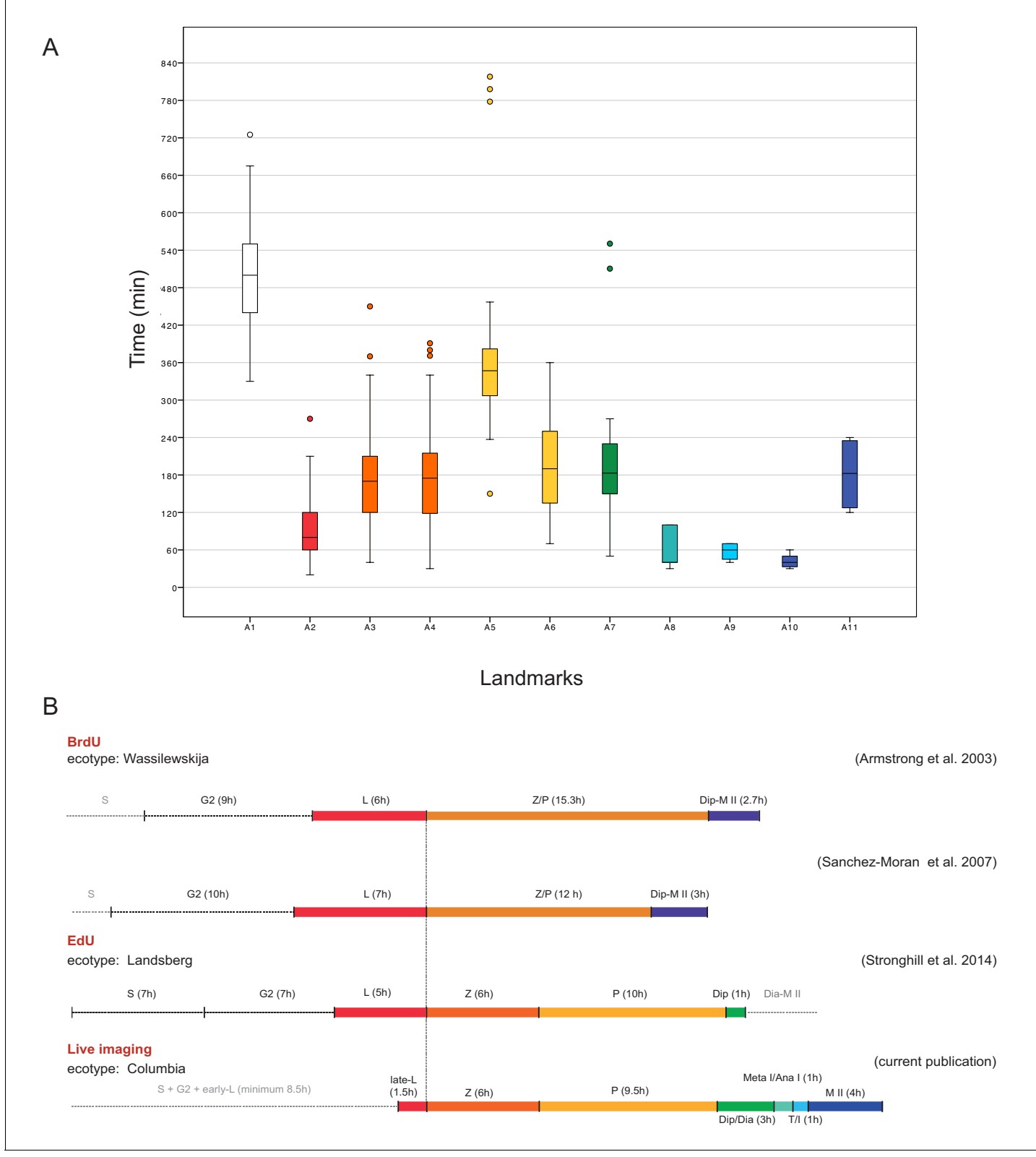

**Figure 7.** Time course of male meiosis in Arabidopsis. (**A**) Box plot illustrating the duration of each landmark in minutes as observed in WT plants. Outliers are illustrated with a dot. The color code for each landmark refers to the meiotic phase: white (A1) is S-phase/G2, red (A2) is late leptotene, orange (A3, A4) is zygotene, yellow (A5, A6) is pachytene, green (A7) is diplotene/diakinesis, aquamarine (A8) is metaphase I/anaphase I, light blue (A9) is telophase/interkinesis, dark blue (A10, A11) is the second meiotic division. (**B**) Comparison of meiotic timelines obtained with different

*Figure 7 continued on next page*

*Figure 7 continued*

techniques: BrdU and EdU staining, followed by sample fixation, versus live imaging. S stands for S-phase; L for leptotene, Z for zygotene, P for pachytene, Dip for diplotene, Dia for diakinesis, Meta I/Ana I for metaphase and anaphase I, T/I for telophase and interkinesis, M II for second meiotic division. The duration of each phase is indicated in hours for all the time courses. Since the onset of zygotene can be clearly defined by previous experiments and in our live cell imaging system, it has been used here to graphically anchor this analysis.

DOI: https://doi.org/10.7554/eLife.42834.019

The following source data is available for figure 7:

**Source data 1.** Table summarizing the number of samples used per each analysis and the duration of each landmark.

DOI: https://doi.org/10.7554/eLife.42834.020

**Source data 2.** Landmark duration in WT.

DOI: https://doi.org/10.7554/eLife.42834.021

A straightforward way to assess the duration of meiosis by our live cell imaging system is by evaluating long movies spanning an entire meiosis. However, long movies with more than 30 hr containing all meiotic stages could only occasionally be obtained and were rarely fully informative due to loss of the focal plane by sample growth. In contrast, 58 movies captured only subsections of meiosis, yet combined provided a complete coverage of meiosis I and II containing each landmark at least 4 times (*Figure 7—source data 1*). To faithfully judge the duration of each landmark, the length of one movie had to be long enough to capture at least two transitions of two sequential landmarks in one individual meiocyte (*Video 2* and *Figure 7—source data 1*). The transition states between two landmarks were added to the observed time of the preceding landmark. Hence, the duration of diakinesis is included in A7, anaphase I in A8, prophase II in A9 and anaphase II in A10 (*Figure 5*). Since we could not faithfully determined S-phase, the transitions between S- and G2-phase was excluded in our time estimates. We then tracked single meiocytes over time with up to 18 meiocytes per anther.

Our measurements of the meiotic phase lengths over all delivered a similar time frame as seen by the previous pulse-chase experiments and we determined the duration of meiosis from late leptotene till telophase II to be 26 hr (*Figure 7*). This value excludes the length of landmark A1 (8.5 hr in total), which is marked by the onset of *REC8* expression, since this time point is not clearly defined with respect to beginning of S-phase. Prophase I, as expected, resulted to be the longest phase (minimum 20 hr) with late leptotene (A2) lasting 1.5 hr, zygotene (A3-A4) 6 hr, pachytene (A5-A6) 9.5 hr and diplotene and diakinesis (A7) together 3 hr. Importantly, we could also resolve meiotic phases thereafter and determine metaphase I and anaphase I (A8) together with 1 hr, telophase I, interkinesis and prophase II (A9) with 1 hr and meiosis II (A10-A11) all together with 4 hr (*Figure 7*, *Figure 7—source data 1* and *2*).

Summing up, the here-presented landmark system allows a dissection of meiosis with unprecedented temporal resolution. Given that the overall length of meiosis as well as the evaluation of individual sub-phases match previously determined durations, we conclude that our imaging system does not perturb meiosis and hence can be applied to analyze different mutants and to assess environmental conditions in the future.

## A case study – analysis of *tam* mutants

To test whether our imaging system can help to promote our understanding of meiotic mutants, we decided to analyze *tam*, which is one of the most studied meiotic mutants in Arabidopsis with at least six published reports focusing on its function (*Bulankova et al., 2013*; *Bulankova et al., 2010*; *Cromer et al., 2012*; *d'Erfurth et al., 2010*; *Magnard et al., 2001*; *Wang et al., 2004a*). What has been reported for *tam* null mutants is that their meiotic progression is delayed in prophase I from pachytene onwards and that they eventually terminate meiosis after the first division. This termination is especially prominent on the male side with nearly 100% of all meiocytes producing dyads instead of tetrads.

We introduced the *KINGBIRD* construct into *tam* mutants and subjected the resulting plants to our live cell imaging procedure. A total of 31 movies capturing 143 male meiocytes were generated covering the entire meiosis in *tam*. We first asked which states of the five cellular parameters can be found in mutant plants and annotated the cellular states of 62 meiocytes. The same states for

nucleolus position and chromatin condensation were found in *tam* in comparison to the WT. When looking into cell shape states, we found neither the triangular nor the tetrad configuration that are characteristic for meiosis II, consistent with the finding that *tam* terminates meiosis after the first meiotic division. Matching a premature termination, we also noted that MT and nucleus position states of meiosis II are absent in *tam*. Strikingly, we discovered an additional state with aberrant MT configurations (state 6), which has not been recognized in previous analyses of *tam* mutants (*Figure 3A*). This state is characterized by the formation of ectopic spindle-like or phragmoplast-like structures in the cytoplasm of meiocytes during diplotene, that is when chromosomes are already connected by chiasmata, but before nuclear envelope breakdown. The ectopic MT configurations were found to adopt different conformations, that is one large array on one side of the nucleus, two separate entities on one side of the nucleus or different clusters of MTs surrounding the nucleus (*Figure 8*). As an additional feature, we also observed, albeit rarely, small dark areas in the cytoplasm occurring already before nuclear envelope breakdown but clearly visible after telophase I, possibly indicating micronuclei (*Figure 8* and *Video 4* and *Video 5*).

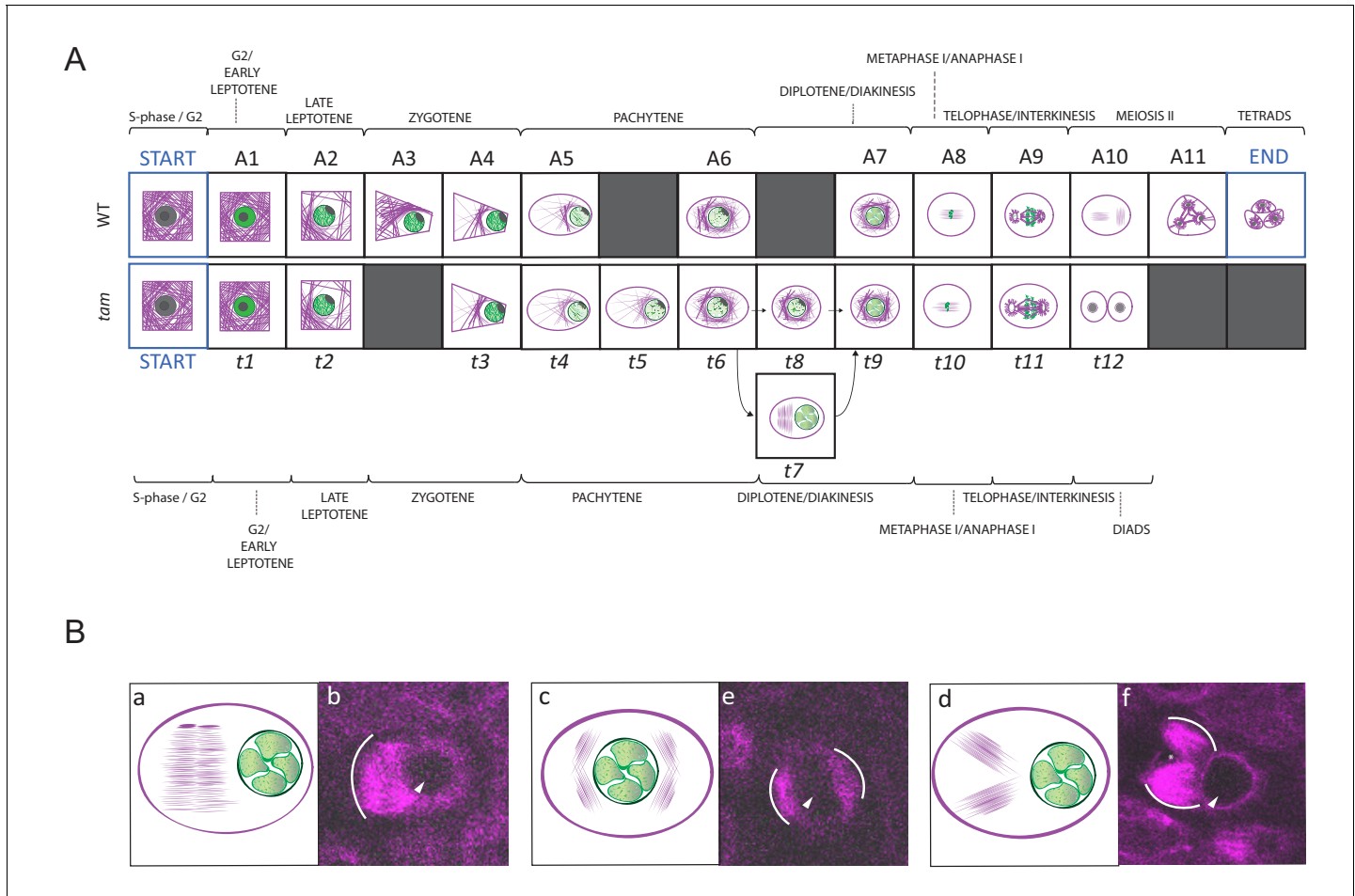

**Figure 8.** Landmark scheme of *tam* and MT aberrations in *tam*. (**A**) Comparison between the landmarks identified for the meiotic progression in WT (A1 to A11) and *tam* (t1 to t11). Corresponding landmarks are located on the top of each other. The gray squares indicate landmarks that are observed in one genetic background only (e.g. A3 and A11 for WT, t5, t7 and t8 for *tam*). Landmark A9 and t11 differ in cell shape while A10 and t12, which follow A9 and t11 respectively mark the time point of premature exit from the meiotic program in *tam* mutants. The arrows connecting t6 to t9 via t7 or t8 show two different pathways seen in *tam* mutants during diplotene. (**B**) Schematic illustrations (a to c) and confocal pictures (d to f) of different patterns adopted by MTs at landmark t7 in *tam* mutants. MTs can bundle in a large phragmoplast-like structure, which develops on one side of the nucleus (a and d). Alternatively they can aggregate in multiple bundles on two opposite sides of the nucleus (b and e) or on the same side of the nucleus (c and f). In the microscopic pictures, the white arc marks the outer rim of the MT aggregations; the arrowhead points to the nucleus, and the asterisk signals the presence of a potential micronucleus.

DOI: https://doi.org/10.7554/eLife.42834.022

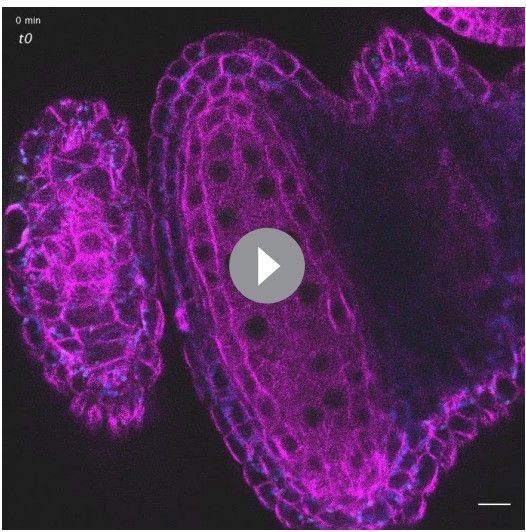

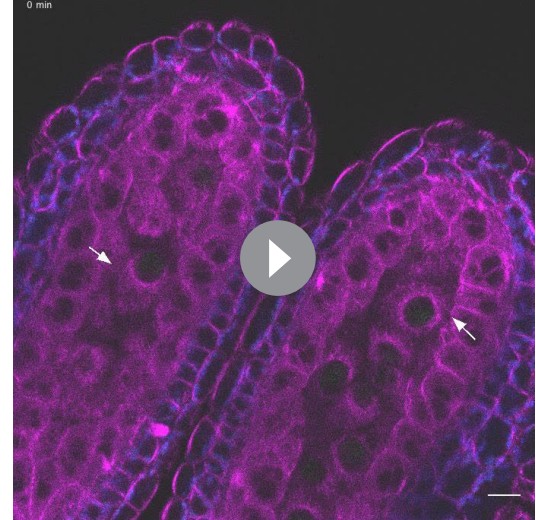

**Video 4.** Complete meiotic progression in *tam*. Progression of meiosis in a *tam* anther. Tubulin (RFP) is highlighted in magenta, chromosomes are marked by REC8 in green (GFP), chloroplasts (autofluorescence) are blue. At the start of the time-lapse meiocytes are at the pre-meiotic landmark *t0* (START), and they proceed through the division, until telophase I (minute 1800 to 2040, *t11*). After telophase I, they exit meiosis prematurely, leading to the formation of dyads (minute 2040 onwards). Meiosis proceed slower in *tam* mutant, starting from diplotene stage, where the MTs start to present aberrant phenotypes (time 1300). On the top left corner, there is an indication of the landmark crossed. Time is expressed in minutes; the interval between image acquisition is 10 min, with the exception of 1 time point (80 min interval between time 880 and 960); time 0 corresponds to the start of image acquisition, and not to the start of meiosis.

DOI: https://doi.org/10.7554/eLife.42834.023

**Video 5.** Aberrant phenotypes in *tam* anthers. Late prophase (from diplotene onwards) in two anthers of a *tam* flower. Tubulin (RFP) is highlighted in magenta, chromosomes are marked by REC8 in green (GFP), chloroplasts (autofluorescence) are blue. Two different cell populations can be recognized: a first population that present MT ectopic aggregations (white arrows) and eventually the formation of micronuclei (white asterisk), and a second population of cells that proceed through the phases in a WT-like manner, albeit slower. Clear premature exit form meiosis is visible in all the cells (starting form time 640). On the top left corner, there is an indication of the landmark crossed. Time is expressed in minutes; the interval between image acquisition is 10 min, time 0 corresponds to the start of image acquisition.

DOI: https://doi.org/10.7554/eLife.42834.024

All together, 102 cellular states were extracted from the movies of *tam* and we next checked the binary co-occurrence of these states (*Figure 3B*). The heatmap reveals a higher degree of disorder for some of these states when compared with the WT situation (highlighted in magenta in *Figure 3C*). Especially prominent was an altered relationship between cell shape and MT array underlining that TAM might regulate MT organization. Likewise, the correlation between chromatin and MT array as well as between nucleus position and cell shape was less stringent in *tam* when compared to the WT. Possibly, the latter difference was induced by the appearance of the aberrant MT structures, which might push the nucleus back to the center of the cell in some cases whereas in the WT the nucleus is strictly positioned at one side of the cell. In turn, this might also lead to cell shape changes in *tam* causing an oval shape again at a time point when in the WT the cell shape has reached a round shape (*Figure 3B*).

Next, we performed our neighboring analysis, coupled with a determination of the time course of meiosis in *tam,* as performed before for the WT (*Figures 4*, *8* and *9*, *Video 4* and *Figure 4—source data 3*, *Figure 7—source data 1*, *Figure 9—source data 1* and *2*). The analysis revealed a total of 12 landmarks in *tam*, named *t1* to *t12* (*Figures 4* and *8*, *Figure 4—source data 3*). The initial cellular stage (1-1-1-1-1) was added in the scheme and named START correspondingly to what was done for the analysis of WT meiosis. As expected, landmarks describing meiosis II were never observed in male meiosis of *tam* mutant plants while new landmarks appeared, that is *t5*, *t7*, and *t8* (*Figure 4*, *Figure 8A*, *Figure 9A*, and *Video 5*). Notably, transition states (non-landmark states colored from

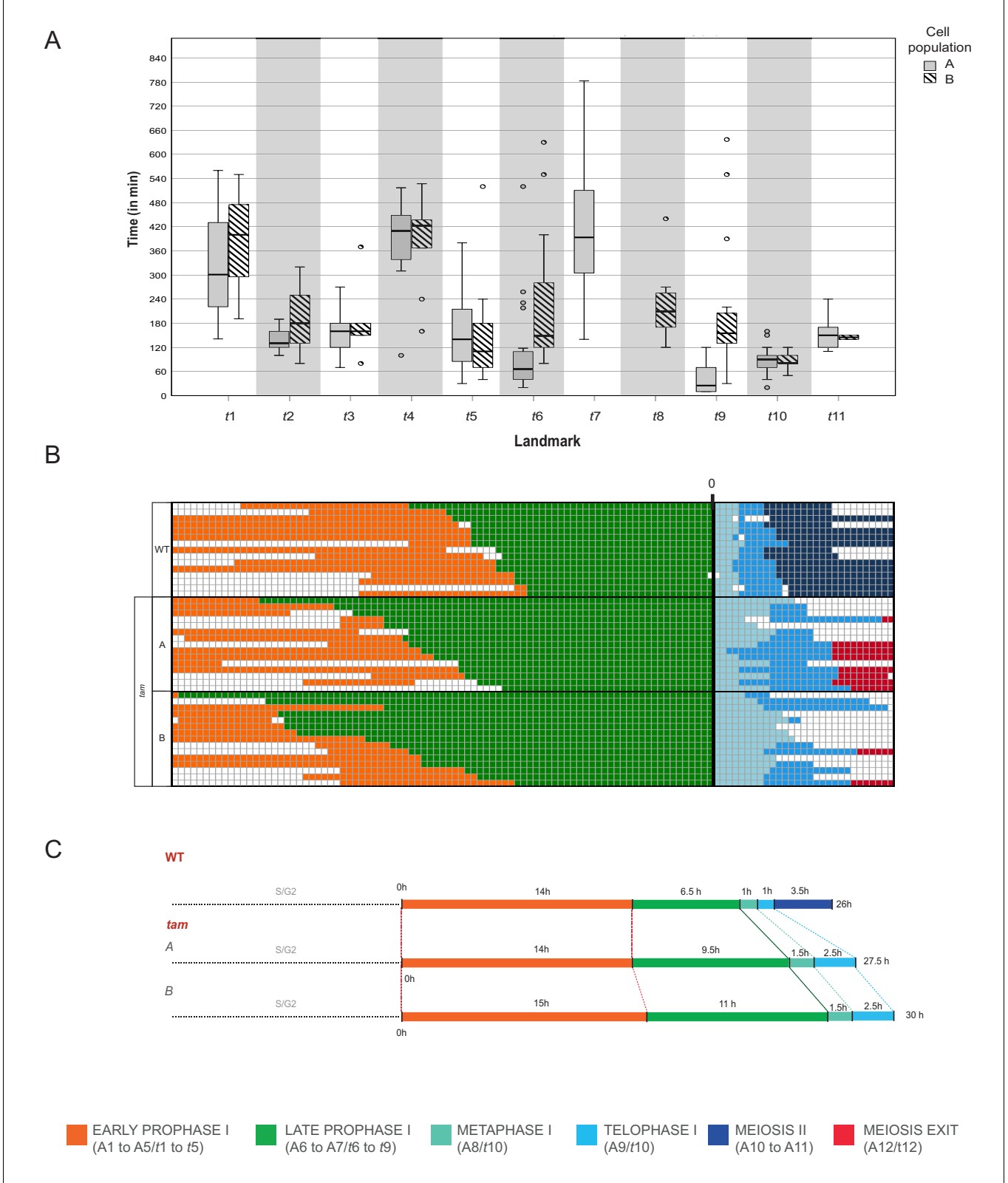

**Figure 9.** Time course of male meiosis in *tam* mutants. (**A**) Box plot illustrating the duration of each landmark in minutes in the two populations of *tam*. Population A follows the pathway including landmark *t7* but skipping *t8* while population B switches from *t6* to *t8* and omits *t7*. Outliers are illustrated with a dot. (**B**) Schematic representation of the progression of meiosis from diplotene onwards in a subgroup of 45 cells (15 WT, 15 *tam* A, and 15 *tam* B) where each line represents a cell. The timing was synchronized at the onset of metaphase (A8 for WT, and *t10* for *tam*), and each square

*Figure 9 continued on next page*

*Figure 9 continued*

corresponds to a 10 min time interval. The graph visualizes that *tam* shows a longer diplotene (green), as well as a prolonged metaphase I and telophase I (aquamarine and light blue) in comparison to the WT. No landmarks for the second meiotic division (dark blue) were observed in *tam*, in which cells exit meiosis (red) directly after telophase I. (C) Comparison of meiotic timelines of WT and *tam*. The time displayed is derived from the sum of the average durations of each single landmark as indicated below the graph.

DOI: https://doi.org/10.7554/eLife.42834.025

The following source data is available for figure 9:

**Source data 1.** Landmark duration of population A, tam.
DOI: https://doi.org/10.7554/eLife.42834.026
**Source data 2.** Landmark duration of population B, tam.
DOI: https://doi.org/10.7554/eLife.42834.027

blue to light yellow in *Figure 4*), reflecting cellular variation, become abundant during the late prophase in *tam* exactly when the here discovered new aspects of the mutant appear.

Prior to the defects in late prophase, we already saw that the wild-type landmark A3 (2-2-2-3-3) is strongly underrepresented in *tam*, that is it is recorded only once out of 6092 total analyzed time points (0.02%) while in WT, A3 is scored 497 times with a percentage of 4.6% and neighboring score pair to 1.32. Consequently, this state obtains a very low neighboring score of −0.49 and it is not a landmark in *tam* (*Figure 4—source data 3*) . The reason for the disappearance of the landmark corresponding to wild-type A3 in *tam* is not entirely clear. Likely linked to the disappearance of this landmark is the extension of landmark *t2* (corresponding to A2 in the WT) suggesting that TAM is required to organize the MT is one corner of the cell. Once this is achieved, *tam* mutant cells appear to quickly change their cell shape and proceed to state *t3* (corresponding to A4 in the WT). Since *t2* takes longer than A2 but then A3 is skipped, the total length of early prophase is very similar between the WT and *tam* (*Figure 9*).

In contrast, late prophase is extended in *tam* (*Figure 9*, *Video 4* and *Video 5*). This prolongation is linked to the appearance of the additional landmark *t5* between the WT landmarks A5 and A6, as well as to the presence of the new landmarks *t7* and *t8* both between the wild-type landmarks A6 and A7.

When looking into the defects of late prophase I in *tam* in detail, we recognized the co-existence of two different populations (A and B) of meiocytes within the same anther. Population A (45.8% of cells, n = 155), develops the above-described spindle-like and phragmoplast-like structure in the cytoplasm while the nucleus is still intact, and proceeds from landmark *t6* via *t7* to *t9*, skipping landmark *t8*. In contrast, population B progresses through landmarks which largely resembled the ones seen in the WT, going from *t6* via *t8* to *t9*, that is skipping landmark *t7*, yet progressing through meiosis with a much slower speed than the WT (*Figures 8A* and *9A*, *Video 4* and *Video 5*). The reason for the appearance for the two populations is as yet not clear and they have probably not been recognized before since population A also is slower that the WT. Thus, *tam* meiocytes proceed through meiosis at a similar speed (*Figure 9*), and do not show strong asynchrony in one anther prior to telophase (*Video 4* and *Video 5*).

Taken together, our analysis assigns TAM the function of a major regulator of MT organization in meiosis. Likely, TAM acts already early in prophase I as seen by the extension of *t2*/A2 in leptotene. One major function appears to be then the repression of premature spindle and/or phragmoplast formation in diplotene.

## Discussion

Live cell imaging has promoted the understanding of many developmental and physiological processes. Examples from plants include live observation of double fertilization (*Hamamura et al., 2014*), following the division pattern in the shoot apical meristem (*Gruel et al., 2016*), the formation of leaf hairs (*Bramsiepe et al., 2010*), and tracking of cell death in the root (*Fendrych et al., 2014*). However, in contrast to following mitotic divisions and subcellular processes in the epidermis, live cell imaging of plant meiosis has been limited (*Ingouff et al., 2017*; *Nannas et al., 2016*; *Sheehan and Pawlowski, 2009*; *Yu et al., 1997*). The reasons for this are foremost the small number of meiocytes and their subepidermal position. In addition, appropriate reporters need to be

introduced into the organism to be analyzed. Here, we have developed a robust live cell imaging system for Arabidopsis male meiocytes based on conventional laser scanning microscopy. We show that this system does not lead to obvious alterations of meiotic progression when compared with previous time course analyses relying on pulse-chase experiments. Notably, our live imaging approach allows individual meiocytes to be followed; this overcomes the problem of asynchrony that occurs in late meiosis, which is likely the reason why the phases from diplotene to the end of meiosis II could not be resolved in the previous time course experiments. Along similar lines, our system allowed the identification of different cell populations in *tam*, which is barely possible without a live cell imaging system.

## A landmark system

Our meiotic description is based on five morphological criteria of male meiocytes that we could distinguish with our reporter genes, that is, cell shape, position of the nucleus, position of the nucleolus, REC8 status and information about chromatin state, and MT array. Importantly, we found that these cellular parameters have two aspects, which make them suitable for a classification system. First, they change in the course of meiosis in a unidirectional manner, for example, cell shape changes from rectangular over trapezoidal and oval to circular. We never found an example where a WT meiocyte skipped one of these cell shape changes or changed back from a later stage to an earlier stage. Second, these parameters are linked with each other and build a matrix. For instance, *full-moon* MT array was never found to be associated with a rectangular cell shape of the meiocyte (*Figure 3B*).

Our analysis of cellular parameters allowed us to identify 11 prominent morphological states, called landmarks A1-A11. These differ from each other by at least one parameter state, and always occur in the same order in any cell progressing through meiosis. The pathway taken by an individual meiocyte to reach each landmark could differ slightly, presumably due to biological variation, and is described by a network of the transition states (*Figure 4*). It is an interesting question to what degree this developmental plasticity depends on meiotic genes and/or environmental factors such as temperature.

The 11 landmarks together with their transitions could be assigned to the classical phases of meiosis (*Figure 5*). However, it has to be noted that the alignment of our landmarks with the classically defined stages remains fuzzy for certain phases. For example, leptotene is defined by the beginning of the chromosome pairing process, with the appearance of the first thin threads, a cell feature that we could not clearly resolve in our analysis. However, as more meiotic reporter lines are generated, for instance for the lateral or central elements of the synaptonemal complex, pairing and synapsis will be resolved with enhanced resolution in future. In this regard, the landmark system is highly modular and expandable depending on the resolution needed by the researcher.

Already with the current setup, our system allows an accurate and robust determination of meiotic stages. This is important since not all cell characteristics can always be unambiguously resolved, for example when the fluorescent signal diminishes because of photobleaching. Hence, the combined parameter states together with the knowledge about the previous cell stages maximize the information gained.

Our landmark system provides a powerful novel platform to study meiocyte differentiation and quantify meiotic progression. The observation that some of the cellular parameters are connected possibly indicates a common regulatory base and/or regulatory dependency. While some associations were expected, for example changes in MT cytoskeleton and cell shape, and are possibly directly linked, other combinations are new and unexpected, for instance the correlation between nucleolus movement and the MT cytoskeleton. These correlations can of course be indirect, yet exploring these combinations in future and identifying which genetic factors underlie them opens a new perspective into meiosis. In turn, their potential uncoupling provides additional, qualitative criteria to describe meiotic mutants.

By observing more features in the future through the use of additional reporter lines and the analysis of mutants affecting meiosis, it will be possible to obtain a highly informative network of functional relationships, that is coupled and uncoupled parameters, within a meiocyte thus heading towards a system-biology understanding of meiosis. Importantly, it will be interesting to see to what degree these cellular parameters can be found and are coupled with each other in female meiocytes.

Similarities and differences can further be compared with the behavior of meiocytes in other plant species.

## A new view onto the function of TAM

Applying our technique to *tam*, a long-known and well-described meiotic mutant in Arabidopsis, revealed surprising new phenotypes and allowed us to quantitatively dissect this mutant. Most strikingly, we observed the formation of spindle or phragmoplast-like structures in the cytoplasm of meiocytes in diplotene, that is at a timepoint when chromosomes are still enclosed in the nucleus. Thus, spindle- and phragmoplast formation can be uncoupled from the presence of chromosomes. However, as soon as the nuclear envelope breaks down and chromosomes are accessible, the ectopic spindle-/phragmoplast-like structures are quickly disassembled and reorganized into a proper meiotic spindle consistent with the finding that chromosomes have a strong MT organizing force. We therefore conclude that one of the major functions of TAM is to prevent self-organization of MTs prior to the presence of highly condensed chromosomes.

These results possibly resemble a situation found in mitotic cells where high CDK activity inhibits the function of NACK1, a kinesin, and NPK1, a MAP3K. NACK1 together with NPK1 trigger a MAP kinase phosphorylation cascade that results in the activation of MAP65-3, a central MT organizing force that drives phragmoplast formation (*Sasabe et al., 2011*). It is tempting to speculate that TAM functions in addition to prevent cytokinesis after the separation of chromosomes in anaphase I in a similar manner. However, the targets of a meiotic CDK-TAM complex are not known and it remains to be seen whether a meiotic NACK1 and/or NPK1 homolog is subject to phospho-regulation by a CDK-TAM complex.

The second major finding when analyzing *tam* mutants was that there are two different populations of mutant meiocytes with only one of these populations developing ectopic spindle- and phragmoplastlike structures. The other population progressed through meiosis reaching landmarks that are also found in the WT, yet in a much slower fashion. This could indicate a crucial dose-dependency for kinase activity, that is cells that for unknown, stochastic reasons have very little kinase activity in addition to the loss of TAM develop ectopic spindle- and phragmoplast like structures while intermediate levels of kinase activity due to the loss of TAM are only slowed down. Alternatively, high kinase activity in diplotene might be needed to establish a special state, for example a spatial mark. In absence of TAM, the establishment of the state is less stable and causes the formation of ectopic spindle- and phragmoplast like structures in these cells. Clearly, additional work is required to unravel the complexity of TAM action in meiosis. However, the here unraveled phenotypes give a clear direction for future experiments and underline the power of life cell imaging without which the behavior of different populations of meiocytes is hardly possible to identify.

# Materials and methods

**Key resources table**

| Reagent type (species) or resource | Designation | Source or reference | Identifiers | Additional information |
|---|---|---|---|---|
| Gene (*Arabidopsis thaliana*) | REC8 | PMID: 11706195 | AT5G05490 | |
| Gene (*Arabidopsis thaliana*) | TAM | PMID: 10072401 | AT1G77390 | |
| Strain background (*Arabidopsis thaliana*) | WT; Wild-Type; Col0 | NASC | Nasc stock number: N1093 | |
| Genetic reagent (*Arabidopsis thaliana*) | *rec8* | Syngenta via NASC | SAIL_807_B08; Syngenta stock name: CS874380 | |

*Continued on next page*

*Continued*

| Reagent type (species) or resource | Designation | Source or reference | Identifiers | Additional information |
|---|---|---|---|---|
| Genetic reagent (*Arabidopsis thaliana*) | *tam* | Syngenta via NASC | SAIL_505_C06; Syngenta stock name: CS836037, Nasc stock number: N836037 | |
| Genetic reagent (*Arabidopsis thaliana*) | *PRO_REC8:REC8:GFP* | this paper | | plant line, Dep. Developemental biology., Hamburg Universitaet, Hamburg, Germany |
| Genetic reagent (*Arabidopsis thaliana*) | *PRO_RPS5A: TagRFP:TUB4* | this paper | | plant line, Dr. Takashi Ishida, Kumamoto University, Japan |
| Genetic reagent (*Arabidopsis thaliana*) | *PRO_RPS5A: TagRFP:TUA5* | this paper | | plant line, Dep. Developemental biology., Hamburg Universitaet, Hamburg, Germany |
| Genetic reagent (*Arabidopsis thaliana*) | *PRO_REC8: REC8:GFP/ PRORPS5A: TagRFP:TUA5* | this paper | | plant line, Dep. Developemental biology., Hamburg Universitaet, Hamburg, Germany |
| Genetic reagent (*Arabidopsis thaliana*) | WT; KINGBIRD WT | this paper | | plant line, Dep. Developemental biology., Hamburg Universitaet, Hamburg, Germany |
| Genetic reagent (*Arabidopsis thaliana*) | *tam*; KINGBIRD *tam* | this paper | | plant line, Dep. Developemental biology., Hamburg Universitaet, Hamburg, Germany |
| Software | Metamorph version 7.8.0 (Molecular Devices) | Molecular devices | | Copyright 1992–2013 Molecular Devices, LLC. |
| Software | Fiji | PMID: 22743772 | https://imagej. net/Fiji | |
| Software | StackReg (Fiji plugin) | DOI: 10.1109/83. 650848 | http://bigwww. epfl.ch/thevenaz/ stackreg/ | |
| Software | Landmark analysis (Phyton script) | this paper | https://gitlab.com/ wurssb/arabidopsis-thaliana—landmark-analysis | Rik Van Rosmalen, Wageningen University and Research, Wageningen, The Netherlands |
| Software | Landmark Summary Generator | this paper | https://github.com/ felixseifert/Landmark SummaryGenerator | Dr. Felix Seifert, cropSeq bioinformatics, Hamburg, Germany |

All materials generated in this work, especially vectors containing plant expression constructs and seeds of the KINGBRID reporter line, are freely available upon request. Requests should be addressed to the corresponding author.

## Plant material and growth conditions

The *Arabidopsis thaliana* plants used in this study were all derived from the Columbia (Col-0) eco-type. The *REC8* T-DNA insertion line *rec8* (At5g05490, SAIL_807_B08) and the *TAM* T-DNA insertion line *tam* (At1g77390, SAIL_505_C06) were obtained from Syngenta via NASC. All genotypes were determined by polymerase chain reaction (PCR) using the primers indicated in *Table 1*. All seeds

**Table 1.** Primers used in this study.

| Purpose | Primer name | Sequence (5′–3′) |
| --- | --- | --- |
| Genotyping | | |
| *rec8 T_DNA* | SAIL_LB3 | TAGCATCTGAATTTCATAACCAATCTCGATACAC |
| | SAIL_807_B08-RP | GGGGGAAAAGAGAAAGGTTC |
| *rec8 WT allele* | SAIL_807_B08-LP | CTCATATTCACGGTGCTCCC |
| | SAIL_807_B08-RP | GGGGGAAAAGAGAAAGGTTC |
| *rec8 WT allelle in REC8:GFP* | SAIL_807_B08-RP | GGGGGAAAAGAGAAAGGTTC |
| | TL-gREC8-R | GAACGGAGAAGGGTAAGGCTCTTGAGTC |
| *tam T_DNA* | SAIL_LB3 | TAGCATCTGAATTTCATAACCAATCTCGATACAC |
| | TAM_L | CAGAAATCCTCCACTTGCG |
| *tam WT allele* | TAM_U | GACTTGATGGATCCACAGC |
| | TAM_L | CAGAAATCCTCCACTTGCG |
| Cloning of PRO$_{REC8}$:REC8:GFP | | |
| REC8 genome | AT5G05490-F | CACCCCAGCCAAGACATTGTGATCTTCAAC |
| REC8 genome | AT5G05490-R | TGTGTGATTCAGGGGTAAGAAATATGCG |
| SmaI | REC8 CterSmaI-F | GGGTAAGGTTTGATTTCTAAATTA |
| SmaI | REC8 CterSmaI-R | GGGCATGTTGGGTCCTCTTGCAAT |
| locus of insertion of gREC8-GFP | gREC8-GFP_LP | GAATATTACCTTGCCATAGGCTTG |
| | attB1r REC8ter-R | GGGGACTGCTTTTTTGTACAAACTTGTGTGTGATTCAGGGTAAGAAA |
| | attB4 REC8_2ndI-F | GGGGACAACTTTGTATAGAAAAGTTAATCAACTCAATTCCCTGTG |
| | attB1r REC8_2ndI-R | GGGGACTGCTTTTTTGTACAAACTTGGCAAAGAGATAAAACCACGC |
| | REC8 2nd intron-F | GCCGCCCCCTTCACCGTAATCAACTCAATTCCCTG |
| | REC8 2nd intron-R | TTCGAATTCCGTTACCTGCAAAGAGATAAAACCAC |
| | Vector for REC8 2I-F | CTCTTTGCAGGTAACGGAATTCGAAATTTA |
| | Vector for REC8 2I-R | AGTTGATTACGGTGAAGGGGGCGGCCGCGG |
| Cloning of PRO$_{REC8}$:REC8:GFP/TagRFP:PRO$_{RPSA5}$:TUA5 | | |
| | attB4 TUA5-F | GGGGACAACTTTGTATAGAAAAGTTTTGATTCGCTATTTGCAGTGCAC |
| | attB1r TUA5-R | GGGGACTGCTTTTTTGTACAAACTTGTGTGTGATTCAGGGGTAAGAAA |

DOI: https://doi.org/10.7554/eLife.42834.028

were surface-sterilized with chlorine gas, sown on 1% agar plates (half-strength Murashige and Skoog (MS) salts, 1% sucrose, pH 5.8) and stored 3 days at 4°C in the dark for stratification. Antibiotics were added for seed selection when required. For germination, plates were transferred to long-day condition (16 h day/8 hr night regime at 22°C/18°C). After germination, plants were transferred to soil and grown under short-day conditions for 2 weeks (12 h day/12 hr night regime at 21°C/18°C), and then transferred to long-day conditions until seed production. For all crosses, flowers of the female parent were emasculated 1 day before anthesis and hand-pollinated 1 to 2 days later.

## Expression constructs: cloning and line selection

To generate the *PRO$_{REC8}$:REC8:GFP* construct, a 7,145 bp genomic fragment of the *REC8* gene containing a 1.8 kbp fragment upstream of the start codon (ATG) and 0.5 kbp fragment downstream of the stop codon was amplified with the primers AT5G05490-F and AT5G05490-R (*Table 1*) and cloned into *pENTR/D-TOPO*. A *SmaI* site was inserted in front of the stop codon of the *REC8* construct by PCR using the primers REC8 CterSmaI-F and REC8 CterSmaI-R (*Table 1*). The ORF for monomeric *GFP* (*mGFP*) was inserted into the *SmaI* site to create *pENTR/PROREC$_{REC8}$:REC8:GFP*, followed by LR recombination reaction into the destination vector *pGWB501* (*Nakagawa et al., 2007*).

A *REC8* reporter line was established by floral dip transformation of *rec8* heterozygous plants with the above-described construct followed by selection of T1 plants on 0.5X MS agar plates supplemented with 25 mg/L Hygromycin B and 50 mg/L Carbenicillin. T2 seeds from individual T1 plants were germinated on 0.5X MS agar plates supplemented with 25 mg/L Hygromycin. T2 line #3 has been selected as the best performing line in terms of *rec8* rescued phenotype and fluorescence intensity.

The $PRO_{RPS5A}$:*TagRFP:TUB4* line has been provided by Takashi Ishida, (Kumamoto University).

A $PRO_{REC8}$:*REC8:GFP*/$PRO_{RPS5A}$:*TagRFP:TUA5* was generated using the MultiSite Gateway. The $PRO_{RPS5A}$:*TagRFP:TUA5* part was amplified from *pGWB501*/$PRO_{RPS5A}$:*TagRFP:TUA5* with the primers attB4 TUA5-F and attB1r TUA5-R and cloned into *pDONR-P4P1r* to create *pDONR-P4P1r/*$PRO_{RPS5A}$:*TagRFP:TUA5*. The *pENTR/*$PRO_{REC8}$:*REC8:GFP* and *pDONR-P4P1r/*$PRO_{RPS5A}$:*TagRFP:TUA5* were combined into the destination vector *R4pGWB501* by LR recombination reaction.

The *KINGBIRD* reporter line in the WT background has been generated via crossing of plants containing the $PRO_{REC8}$:*REC8:GFP* and $PRO_{RPS5A}$:*TagRFP:TUB4* constructs as described above. The *KINGBIRD* reporter line in the *tam* background was generated via Agrobacterium-mediated transformation of the vector $PRO_{REC8}$:*REC8:GFP*/$PRORPS_{RPS5A}$:*TagRFP:TUA5* into heterozygous *tam* plants.

## Phenotype evaluation

Rescue of the *rec8* phenotype was assessed at pollen level using a Peterson staining protocol as described in *Peterson et al. (2010)* and monitoring meiotic progression at a cytological level via cell spreads as described in *Ross et al. (1996)*.

## Live imaging of meiotic division

Flowers of 0.4–0.6 mm were isolated and prepared as presented in the results section '*Specimen preparation*'. Up to four samples were positioned on the same petri dish and cultured in Arabidopsis Apex Culture Medium (ACM): half-strength Murashige and Skoog (MS) salts, 1% sucrose, 0.8% agarose, pH5.8. Supplements were added to a 1X concentration from a 1000X stock solution (stock solution: 10% Myoinositol, 0.1% nicotinic acid, 0.1% pyridoxine hydrochloride, 0.1% thiamine hydrochloride, 0.2% glycine dissolved in Millipore water and subsequently filter sterilized) (*Hamant et al., 2014*). Time lapses were acquired using a Zeiss LSM 880 confocal microscope and ZEN 2.3 SP1 software (Carl Zeiss AG, Oberkochen, Germany). During image acquisition the petri dish was filled with autoclaved water and placed under a W-plan-Apochromat 40X/1.0 DIC objective (Carl Zeiss AG, Oberkochen, Germany). GFP was excited at λ 488 nm, and detected at λ between 498–550 nm. RFP was excited at λ 561 nm and detected at λ between 578–650 nm. Autofluorescence from chloroplasts was highlighted in blue using excitation at λ 488, and detection at λ between 680–750 nm. Time lapses were acquired as series of Z-stacks (six planes, 50 µm distance). Interval time was varying from a max of 15 to a min of 3 min depending on sample conditions. The functions 'Autofocus' and 'Automatized positions' were used to acquire images. Room temperature and sample temperature were controlled and stabilized at 18°C and 21°C respectively.

## Image processing

First, the time lapses were converted into sequential images. The focal plane was then selected at each time point using the function '*Review Multi Dimensional Data*' of the software Metamorph, version 7.8.0.0. The files were then exported and saved as. tiff. Image drift was corrected by the Stack Reg plugin (Rigid Body option) for Fiji (Fiji version 1.52b, https://imagej.net/Fiji). Cell numbers were assigned manually.

## Quantitative analysis of live cell imaging data: Data set description

The landmark system is based on the analysis of a subset of data on male meiocytes from WT and *tam* plants carrying the KINGBIRD reporter constructs. A subset of the analyzable male meiocytes was described at every time point by assigning manually a value for each of the five parameters assessed. For the WT, a total of 169 meiocytes from 35 anthers were annotated, leading to a total of 18,531 data points spanning 3,269 hr. For 7860 observations one or more of the parameters could not be annotated by a well-defined state, with 5893 observations not having a single parameter recognizable. The resulting dataset, consisting of 10,671 time points, was used to determine the co-

occurrences of parameter states and the landmarks. For *tam* a total of 62 meiocytes from 19 anthers were annotated, leading to a total of 10,224 data points spanning 1,694 hr. For 4127 observations one or more of the parameters could not be annotated by a well-defined state, with 3109 observations not having a single parameter recognizable. The resulting dataset, consisting of 6097 time points, was used to determine the co-occurrences of parameter states and the landmarks.

For the analysis of the single reporters, the same data set derived from the observation of WT meiocytes was used. We subdivided it into two groups: the first group contained the annotations for the two parameters REC8/Chromatin and nucleolus position, while the second group included the annotations of the remaining three parameters: cell shape, nucleus position and MT array. Each group contained the annotations of the same cells at the same time points.

Two additional datasets were annotated using the landmark system directly, that is assigning the cellular states A1-A11 and *t*1-*t*12 respectively and were used for the calculation of the time course of WT and *tam* plants. The landmark attribution for each cell at each time point was done manually. Data were recorded in the CSV format and data analysis was done using the Python programming language (Version 3.6, Python Software Foundation, https://www.python.org; https://gitlab.com/wurssb/arabidopsis-thaliana—landmark-analysis) (*van Rosmalen et al., 2019*).

### Landmark extraction: Data preprocessing

The manually created data set used for landmark extraction contains a description of the state of each of the five parameters (two parameters in the case of $PRO_{REC8}:REC8:GFP$ alone or three for $PRO_{RPS5A}:TagRFP:TUB4$ alone) that were recorded for individual cells at 15 min intervals. Missing data points were labeled by 'n', this was done to ensure that unmeasured periods are noted properly and to avoid assigning any unrealistic transitions. The combination of states of each of the cellular parameters makes up the cellular state. Transitions from one cellular state to another occur when one or more parameters change to a new state.

### Landmark extraction: Cell state co-occurrence

To create the co-occurrence heat map in *Figure 4*, we counted the number of times a combination of two parameter states occurred. Since some time lapses were measured with different temporal resolution (e.g. 10 min intervals versus 15 min intervals), we first resampled the data points from all time lapses to have the same time between measurements. Co-occurrence counts were normalized by the total number of counts of the columns parameter state, including the counts where the state of the 2nd parameter could not be measured.

### Landmark extraction: Bootstrapping

To assess the robustness of the selected landmarks and thus our theoretical framework, we performed a bootstrapping procedure on our data set. The total set of observations was randomly sampled with replacement to obtain a data set 1.5 times the size of the original data set. Scores for each state in this data set were calculated using the procedure described in the previous paragraph and in the section 'A meiotic landmark system' of the results. This process was repeated 1,000 times to obtain estimates for the mean value, standard deviation and quantiles of the score of each cellular state. Results of the bootstrap can be seen in *Figure 4—source data 2* and *4*.

### Meiotic time course calculation

The duration of each landmark was automatically extracted from the CSV files (Material and methods, *Quantitative analysis of live cell imaging data, Data set description*) using custom software based on consecutive landmark transitions (*Seifert, 2019*; copy archived at https://github.com/elifesciences-publications/LandmarkSummaryGenerator). This resulted in a dataset of 327 landmark durations from 136 meiocytes of 17 different anthers for WT plants, and of 245 landmark durations from 76 meiocytes of 15 different anthers for *tam* plants.

## Acknowledgements

We thank Chris Franklin (Birmingham University, UK), Maren Heese (University of Hamburg, DE), and Vanesa Calvo-Baltanas (Wageningen University and Research, NL) for critical reading and helpful

comments to the manuscript. We are grateful to Olivier Hamant (ENS, Lyon) for training in imaging. The Versailles Arabidopsis Stock Center and the Nottingham Arabidopsis Stock Centre (NASC) are acknowledged for providing material used in this study. We thank Takashi Ishida (Kumamoto University, Japan) for providing us with the $PRO_{RPS5A}$:TagRFP:TUB4 construct.

## Additional information

### Competing interests
Felix Seifert: is affiliated with CropSeq bioinformatics. The author has no other competing interests to declare. The other authors declare that no competing interests exist.

### Funding

| Funder | Grant reference number | Author |
| --- | --- | --- |
| European Union 7th Framework Programme | ITN-606956 | Maria A Prusicki<br>Erik Wijnker<br>Arp Schnittger |
| University of Hamburg | Core funding | Maria A Prusicki<br>Shinichiro Komaki<br>Felix Seifert<br>Katja Müller<br>Erik Wijnker<br>Arp Schnittger |

The funders had no role in study design, data collection and interpretation, or the decision to submit the work for publication.

### Author contributions
Maria A Prusicki, Conceptualization, Investigation, Writing—original draft; Emma M Keizer, Rik P van Rosmalen, Software, Visualization, Methodology, Writing—review and editing; Shinichiro Komaki, Methodology, Writing—review and editing; Felix Seifert, Software, Writing—review and editing; Katja Müller, Formal analysis, Investigation, Visualization; Erik Wijnker, Formal analysis, Supervision, Investigation, Visualization; Christian Fleck, Conceptualization, Resources, Supervision, Funding acquisition, Methodology, Writing—original draft, Project administration, Writing—review and editing; Arp Schnittger, Conceptualization, Resources, Supervision, Funding acquisition, Methodology, Writing—original draft, Writing—review and editing

### Author ORCIDs
Maria A Prusicki (iD) https://orcid.org/0000-0003-3755-3402
Shinichiro Komaki (iD) https://orcid.org/0000-0002-1189-288X
Arp Schnittger (iD) https://orcid.org/0000-0001-7067-0091

### Decision letter and Author response
Decision letter https://doi.org/10.7554/eLife.42834.031
Author response https://doi.org/10.7554/eLife.42834.032

## Additional files

### Supplementary files
• Transparent reporting form
DOI: https://doi.org/10.7554/eLife.42834.029

### Data availability
All data generated or analysed during this study are included in the manuscript and supporting files. Custom software used is available at https://github.com/felixseifert/LandmarkSummaryGenerator (copy archived at https://github.com/elifesciences-publications/LandmarkSummaryGenerator).

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
