## [Decision Letter]

Thank you for submitting your article "Live cell imaging of meiosis in *Arabidopsis thaliana* – a landmark system" for consideration by *eLife*. Your article has been reviewed by three peer reviewers, one of whom is a member of our Board of Reviewing Editors, and the evaluation has been overseen by Christian Hardtke as the Senior Editor.

The reviewers have discussed the reviews with one another and the Reviewing Editor has drafted this decision to help you prepare a revised submission.

Summary:

Meiosis is an essential process in the life of plants and animals, but due to it occurring in hidden tissues, it has been recalcitrant to study by live-cell imaging methods. Here Schnittger and co-workers develop an ex vivo anther system, a new live-cell imaging reporter and a framework for classifying the events of meiosis. Overall this is clearly explained, technically impressive and will provide a useful and needed technique. This study contains two important innovations for addressing the live imaging of plant meiosis. First, it establishes fluorescent reporter markers that highlight important cellular features for visualization of meiosis and describes detailed protocol for microscopic analysis of male meiocytes in the context developing flower. Second, by characterizing five cellular parameters and their behavior in the course of meiosis, the authors created a landmark system for describing meiotic progression assessed by live imaging.

Essential revisions:

1) As this work is being put forth as a novel technique that should enable the characterization of mutants or environmental perturbations, an example of the systems application to either one of these situations should be provided. In our reviewer discussions, it became clear that for widespread adoption, people would want evidence that it was capable of reporting deviations from WT before introgressing the reporters into their favorite backgrounds. Is the system robust enough for characterization of meiotic mutants as it is expected that disturbance of meiosis will affect cellular parameters used to define the landmark system?

2) Regarding the general applicability of the landmark system. As it is defined now, it relies on the *KINGBIRD* reporter line that contains both RFP and GFP. This, in fact, precludes utilization of the system for guiding temporal localization of other proteins of interest in the course of meiosis. It would be very useful to determine whether the 11 landmarks (or most of them) can also be detected without including either the REC8 or microtubule markers (simply by doing the analysis without one of these parameters).

3) There seems to be some dependency among state assignments based on other events. For example, in Figure 3, looking at nuclear position state 1 and 3 or 4 and 6 are indistinguishable, so my guess is something was defined as being 1 or 3 based on the results from the other markers. In Figure 5, things are nicely laid out with 11 distinct fates, but could A2 be the from stage 1 or stage 3 nuclear position? Why this issue isn't a problem needs to be explained more clearly in the main text.

---

## [Author Response]

Essential revisions:1) As this work is being put forth as a novel technique that should enable the characterization of mutants or environmental perturbations, an example of the systems application to either one of these situations should be provided. In our reviewer discussions, it became clear that for widespread adoption, people would want evidence that it was capable of reporting deviations from WT before introgressing the reporters into their favorite backgrounds. Is the system robust enough for characterization of meiotic mutants as it is expected that disturbance of meiosis will affect cellular parameters used to define the landmark system?

To address this point, we have decided to revisit mutants in *tardy asynchronous meiosis* (tam), which encodes for CYCLIN A1;2. With at least 6 research publications focusing on this mutant, TAM is one of the best-characterized meiotic genes. The rationale was to challenge our system with the question whether we would find new phenotypic aspects in *tam* plants. Indeed, this was the case. The work is presented now in the subsection “A case study – analysis of *tam* mutants”.

In short, we find a striking new mutant phenotype, i.e. the formation of spindle-like/phragmoplast-like structures in *tam*. These structures are formed prior to nuclear envelope breakdown. Thus, these structures are formed without chromosomes serving a nucleation sites. A similar case has, to our understanding, not been reported in any other organism and experts in the field of microtubules dynamics and spindle formation whom we have consulted in this manner were very excited about these phenotypes. We conclude that TAM coordinates microtubule formation with nuclear events. This in turn offers a fundamentally new explanation why *tam* mutants exit meiosis after the first division, i.e. TAM likely represses cytokinesis by preventing the formation of an early phragmoplast after meiosis I. The power of our live cell imaging system became especially obvious by the finding of two populations of *tam* mutant cells that behave differently. The identification of these two populations would be hardly possible by analyzing fixed cells where the destiny of a single cell cannot be followed. Moreover, we can quantitatively describe the time course of these different *tam* mutant populations. Thus, we think that our system is a very valuable addition to the available toolset for the analysis of meiotic mutants.

2) Regarding the general applicability of the landmark system. As it is defined now, it relies on the KINGBIRD reporter line that contains both RFP and GFP. This, in fact, precludes utilization of the system for guiding temporal localization of other proteins of interest in the course of meiosis. It would be very useful to determine whether the 11 landmarks (or most of them) can also be detected without including either the REC8 or microtubule markers (simply by doing the analysis without one of these parameters).

The reviewers raise an important point here and he have run our analysis with only one reporter, either REC8-GFP or TFP-TUA4. These results are provided in the subsection “Landmarks analysis can be performed with one marker only, yet less informative”.

In short, we find that both reporters deliver landmarks by themselves. As expected, the tubulin reporter provides a greater resolution in absolute terms as it highlights more cellular features. However, the combination with other fluorescently tagged proteins really depends on the research question and for some incidences it might still be interesting to use REC8, i.e. when processes in pachytene need to be analyzed. We also find that three out of eleven landmarks can only be observed through the combination of both the microtubule reporter and REC8 and thus, the use of REC8 is not superfluous. These results are presented in the new Figure 5 and in Figure 5—source data 1.

As an additional possibility, a third color could be introduced in the data acquisition, for example changing GFP to YFP and then adding mTurquoise. This would allow following three proteins at the same time although care has to be taken in case of co-localization due to the slightly overlapping emission spectra.

3) There seems to be some dependency among state assignments based on other events. For example, in Figure 3, looking at nuclear position state 1 and 3 or 4 and 6 are indistinguishable, so my guess is something was defined as being 1 or 3 based on the results from the other markers. In Figure 5, things are nicely laid out with 11 distinct fates, but could A2 be the from stage 1 or stage 3 nuclear position? Why this issue isn't a problem needs to be explained more clearly in the main text.

To clarify this point, the drawing in Figure 3A has been updated with more precise sketches, and the Figure 5 has been updated accordingly. Furthermore, a detailed explanation on the parameters has been added in the main text.